



# Characterising Mass-resolved Mixing State of Black Carbon in Beijing Using a Morphology-Independent Measurement Method

Chenjie Yu[1], Dantong Liu[1,5], Kurtis Broda[2], Rutambhara Joshi[1], Jason Olfert[2], Yele Sun[3], Pingqing Fu[3,6], Hugh Coe[1], James D. Allan[1,4]

[1]School of Earth and Environmental Science, University of Manchester, Manchester, M13 9PL, UK
[2]Department of Mechanical Engineering, University of Alberta, Alberta, T6G 2R3, Canada
[3]Institute of Atmospheric Physics, Chinese Academy of Sciences, Beijing, 100029, China
[4]National Centre for Atmospheric Sciences, University of Manchester, Manchester, M13 9PL, UK
[5]Now at Department of Atmospheric Sciences, School of Earth Sciences, Zhejiang University, Zhejiang, 310027, China
[6]Now at Institute of Surface-Earth System Science, Tianjin University, Tianjin, 300072, China

*Correspondence to*: Dantong Liu (dantongliu@zju.edu.cn), James Allan (james.allan@manchester.ac.uk)

**Abstract.** Refractory Black Carbon (rBC) in the atmosphere is known for its significant impact on the climate system in the atmosphere. The relationship between the microphysical and optical properties of rBC remain uncertain and are largely influenced by the size, coating thickness and mixing state of particles. This study presents a coupling of a centrifugal particle mass analyser (CPMA) and a single particle soot photometer (SP2) for the morphology-independent quantification of the mixing state of rBC-containing particles, used in the urban site of Beijing as part of the Air Pollution and Human Health-Beijing (APHH-Beijing) project during winter ($10^{th}$ Nov – $10^{th}$ Dec) and summer ($18^{th}$ May – $25^{th}$ June). An inversion method is applied to the measurements to present a two-variable distribution of both rBC core mass and total mass of rBC-containing particles and present the mass-resolved mixing state of rBC-containing particles. The mass ratio between non-rBC coating and rBC core (MR) is calculated to determine the coating thickness of the rBC-containing particles. The bulk MR was found to vary between 2 - 12 in winter and between 2 - 3 in summer. This mass-resolved mixing state is used to derive the mixing state index ($\chi$) for the rBC-containing particles. $\chi$ quantifies whether the coating is evenly distributed across the rBC-containing particle population and is used to determine the degree of internal and external mixture of rBC-containing particles. The rBC-containing particles in Beijing were found to be 55% - 70% internally mixed in winter depending on the dominant air masses. $\chi$ of rBC-containing particles was highly positively associated with increased bulk MR, rBC mass loading or pollution level in winter, whereas $\chi$ of rBC-containing particles in summer varied significantly (ranging 60% - 75%) within the narrowly-distributed bulk MR and was found to be independent of air mass sources. This concludes that the bulk MR may only act as a predictor of mixing state in winter, and $\chi$ is better to quantify the mixing state of rBC-containing particles. The same level of bulk MR corresponded with a higher $\chi$ in summer than in winter and this tended to suggest a limited formation of coatings on rBC largely depended on primary sources. However, with the higher Non-refractory $PM_1$ (NR-$PM_1$) concentration in winter, the coagulation process may still lead relative thick coatings. In summer the higher secondary compounds made the rBC-containing particles more homogeneous. But due to the higher temperatures and limited pollution level, the coating thickness in summer is limited. The mixing state of rBC-containing particles should also depend on the coating formation mechanism,



both primary source influence and secondary coating formation mechanism should be considered in interpreting the rBC-containing particles mixing state in the atmosphere. This particle morphology-independent and mass-based data format as introduced in this study could be conviently applied in particle-resolved or other process models to investigate atmospheric rBC aging and mixing state properties.

## 1 Introduction

Black Carbon (BC) is an important light absorbing carbonaceous component of the atmosphere and is regarded as dominant amongst absorbing aerosols in the atmosphere (Bond and Bergstrom, 2006). Atmospheric BC particles have both a direct positive impact and semidirect impact on Earth's radiative balance (Jacobson, 2000; Bond et al., 2013; IPCC, 2013). However, the climate impact of atmospheric BC particles contains large uncertainties (Ramanathan and Carmichael, 2008). It is known that when BC is coated with other compounds, the absorption will be enhanced due to the so-called 'lensing effect' compared to the bare BC (Jacobson, 2001; Lack and Cappa, 2010) and the magnitude of coating thickness has a significant impact on the absorption properties of BC-containing particles (Moffet and Prather, 2009). However, the optical properties of BC-containing particles vary with BC mixing state, morphology, and chemical compounds which induces large uncertainties in the calculation of the BC absorption enhancement (Cappa et al., 2012; Liu et al., 2017). A number of observations (Cappa et al., 2012; Peng et al., 2016; Liu et al., 2017; Zhang et al., 2018b) and model studies (Wu et al., 2018; Zhang et al., 2018a) suggest that the absorption enhancement ranges from 1.0 up to 3.5. Though a number of different proposed model treatments give different predictions of BC mixing state (Jacobson, 2001; Murphy et al., 2006; Péré et al., 2009), none of these give perfect agreement with atmospheric observations, either in situ or remote (Fierce et al., 2016). As the coated BC particles may act as cloud condensation nuclei (CCN) in the atmosphere, the coating thickness information of BC-containing particles can also influence the wet removal rate and thus its atmospheric lifetime and overall climate forcing potential (Kuwata et al., 2007; Shiraiwa et al., 2007). Taylor et al. (2014) shows that the size distribution of BC-containing particles is important for the simulation of BC lifetime, vertical profile and transportation. The results from Liu et al. (2015) also show that the coating thickness of BC particles can largely influence their scavenging efficiency and lifetime in the clouds. The BC mixing state is especially important for polluted environments, where a combination of active sources, mixing and secondary aerosol formation mean that the BC mixing state is highly dynamic (Riemer et al., 2009). The mixing state of BC in most models remains unconstrained due to lack of measurement data, so this study aims to address this need by applying a new measurement technique to the atmosphere and offering new insight into the behaviour of BC in an important megacity.

To clearly quantify how well the BC particles are mixed in the atmosphere, the parameter "mixing state index ($\chi$)" is applied in this study. Inspired by ecology studies (Whittaker, 1960, 1965), Riemer and West (2013) introduced a mathematical framework to quantify the aerosol mixing state. In this framework, the mixing state index ($\chi$) is defined as the ratio between the average single particle species diversity ($D_\alpha$) and the bulk particle species diversity ($D_\gamma$), and the mixing state index ($\chi$)



varies from 0% (full external mixing; *i.e.*, all particles consist of only one species which may be different for each particle) and 100% (full internal mixing; *i.e.*; all particles are equally composed of all species). A number of measurement studies have generated this statistic by reporting a mass fraction of different compositions through various measurement techniques. Healy et al. (2014) reported that the variation of bulk aerosols mixing state index in Paris winter is more related to the dominated air

mass source by applying the single particle aerosol mass spectrometer. Dickau et al. (2016) introduces a size and mass selection method to quantify the volatile mixing state of soot. As well as online measurements, offline microscopy techniques are also being used to retrieve species mass fractions. Bondy et al. (2018) performed a microscopy study on air samples collected from the south-eastern United States, and they found that most of the accumulation mode particles are internally mixed when the secondary organic aerosol concentration is high, and most of them are externally mixed during dust events. As one of the large

mega-cities in China, Beijing has suffered from heavy air pollution (Yang et al., 2005) and is likely to have many factors affecting BC mixing state, which in turn has implications for both air quality and climate (Ching and Kajino, 2018). These aerosols can have a global reach (Liu et al., 2015) and as their effects may be influenced by dynamic effects near the point of emission where concentrations are highest, an polluted urban environment represents a good location to investigate these behaviours and properties.

In this study, a combination of a Centrifugal Particle Mass Analyser (CPMA, Cambustion Ltd., Cambridge, UK) and a single particle soot photometer (SP2; DMT, Boulder, CO, USA) were coupled in the field for the first time and with a new inversion algorithm applied to characterise the mixing state of rBC-containing particles in Beijing. In this context, the BC is referred to as 'refractory black carbon' or rBC as an operational definition, consistent with Petzold et al. (2013). Liu et al. (2017) first

introduced this morphology-independent instrument configuration to measure the different mass ratio between non-rBC material and rBC material. Compared to the traditional SP2 only measurement, the significant improvement of the CPMA-SP2 method is that it can measure the non-rBC coating mass directly which means no assumption is needed for coating density or even core-shell morphology. To clearly characterize the distribution of both rBC core and non-rBC coating material as part of a continuous scan (as opposed to discrete masses used previously by Liu et al. (2017)), the CPMA-SP2 results are presented

in a two variable distribution by applying a novel inversion method presented by Broda et al. (2018). The coating thickness of rBC-containing particles and the coating distribution information across the rBC-containing particle population is derived from the measurement results, and the implication of these results are discussed.

## 2 Experiment details

### 2.1 APHH field campaign

With the aim of identifying the mixing states of rBC in a complex urban environment, an experiment was devised as part of the Air Pollution and Human Health-Beijing (APHH-Beijing) programme. The details of the wider campaign is described in



Shi et al. (2018). The measurements were conducted during the APHH winter (10[th] Nov – 10[th] Dec) and summer (18[th] May – 25[th] June) intensive observation periods at the Institute of Atmospheric Physics (IAP) tower site. The site (39°58′28″N, 116°22′16″E) is located in between the 3[rd] ring and 4[th] ring in the urban area of Beijing to the north of the city centre. As part of the AIRPRO and AIRPOLL joint campaign, these measurements also contribute to the better understanding of the critical pollutants and the pollution process in urban Beijing.

## 2.2 Centrifugal Particle Mass Analyser (CPMA)

The CPMA sorts particles by their mass to charge ratio. The details of the CPMA have been described elsewhere (Olfert and Collings, 2005; Olfert et al., 2006). Briefly, two rotating coaxial cylindrical electrodes rotate at different angular velocities inside the CPMA. Particles of a narrow range of mass to charge ratio will exit the classifier based on a balance between electrostatic and centrifugal forces depending on the voltage difference between the electrodes and their rotational speed. The average mass of the particles ($m$) exiting the CPMA is:

$$m = \frac{\phi e V}{\omega_c^2 r_c^2 \ln\left(\frac{r_o}{r_i}\right)} \quad (1)$$

Where, $e$ is the electronic charge ($1.6 \times 10^{-19}$ C), $\phi$ is number of charges carried by each particle, $V$ is the voltage difference, $r_o$ is the outer cylinder radius, $r_i$ is the inner cylinder radius, and $\omega_c$ is the rotation speed of the gas at the center of the gap between the cylinders $r_c$ ($r_c = \frac{1}{2}(r_o + r_i)$). The charging in this instance was provided by a bipolar electrical neutralizer (MSP Corp., Shoreview, USA) that gives an equivalent charging profile to a traditional radioactive neutralizer for particles larger than 10 nm.

## 2.3 Single Particle Soot Photometer (SP2)

The SP2 is widely used in atmospheric rBC studies (Stephens et al., 2003). It consists of four optical detectors and one Nd:YAG crystal laser with a Gaussian intensity distribution (Schwarz et al., 2010). The SP2 can measure the particles with an optical diameter of 200 nm – 720 nm, and 70-850 nm for rBC-containing particles (Liu et al., 2010; Adachi et al., 2016). The details of the SP2 algorithm in Manchester is described in Liu et al. (2010) and McMeeking et al. (2010). The Manchester SP2 incandescence signal was calibrated using Aquadag black carbon particle standards (Aqueous Deflocculated Acheson Graphite, manufactured by Acheson Inc., USA) before the measurement, and the correction factor is 0.75 for ambient rBC measurement was applied (Laborde et al., 2012).



## 2.4 Manchester CPMA-SP2 system

A novel coupling measurement method, CPMA-SP2, has been applied to measure the non-rBC coating mass among the population of rBC without morphology dependence (Liu et al., 2017). To set up the CPMA-SP2 system, the CPMA is placed upstream of the SP2 to classify particles of a nominal mass, and sequentially pass these monodisperse particles to the SP2 for analysis of the mass of rBC from the individually classified particles. When sampling with the CPMA-SP2 system, the CPMA was set to select particles from 0.3 fg up to ~ 15 fg. The CPMA performed one scan every 30 min, and it took ~20 min to complete one scan. For the remaining time, the CPMA rotation and voltage were disabled, meaning the SP2 was measuring the polydisperse aerosol measurement (a comparison between this and a bypassed configuration demonstrated that this introduced minimal losses). The CPMA-SP2 set up with mono-mass and poly-dispersed scan sequence is shown in Fig. S1.

## 2.5 NR-PM$_1$ (Non-refractory particulate matter with aerodynamic diameter <1 μm) measurement and back trajectory model

A High-Resolution Time-of-flight Aerodyne Mass Spectrometer (HR-ToF-AMS, Aerodyne Research Inc., USA) was used to measure the bulk population of aerosol particle chemical compositions. The details of HR-ToF-AMS is described by DeCarlo et al. (2006). The mass concentration of the NR- PM$_1$ (Non-refractory particulate matter with aerodynamic diameter <1 μm) particle is calculated from the sum of organics, sulphate, nitrate, ammonium and chloride.

The HYSPLIT back trajectory model (Draxler and Hess, 1998) was run with the 1°×1° horizontal and vertical wind fields provided by the GDAS1 reanalysis meteorology data. The HYSPLIT model analysis was used to classify the sources of air mass influenced the measurement site, and this method is described by Liu et al. (2018b). As shown in Figure 1, the continental air mass sources are classified into five regions: Northern Plateau, Southern Plateau, Western North China Plain (NCP) and Eastern NCP.

## 3 Methods

### 3.1 CPMA-SP2 inversion

To quantify the rBC mixing state in Beijing, the CPMA-SP2 results have been applied to a new inversion method introduced by Broda et al. (2018), and the calculation process is described in the supplements. Here a two-variable distribution function is used to describe the distribution rBC mixing states:

$$\frac{\partial^2 N}{\partial log m_{\mathrm{p}} \partial log m_{\mathrm{rBC}}} \qquad (2)$$



where $m_\mathrm{p}$ is the total mass of an individual rBC-containing particle, and $m_\mathrm{rBC}$ is the mass of an individual rBC particle. $\partial^2 N$ represents the number concentration of rBC-containing particles with their total particle mass between $m_\mathrm{p}$ and $m_\mathrm{p} + \mathrm{d}m_\mathrm{p}$ and rBC particle mass between $m_\mathrm{rBC}$ and $m_\mathrm{rBC} + \mathrm{d}m_\mathrm{rBC}$. This two-variable distribution can also be integrated to single-variable

number distributions ($\frac{\mathrm{d}N}{\mathrm{dlog}m_\mathrm{p}}$ and $\frac{\mathrm{d}N}{\mathrm{dlog}m_\mathrm{rBC}}$) that are normally used in aerosol studies. An example of the original two-

dimensional distribution $\frac{\partial^2 N}{\partial \log m_\mathrm{p} \partial \log m_\mathrm{rBC}}$ graph from the measurement is presented in the Figure S2.

### 3.2 Mixing state of rBC-containing particles calculation

Riemer and West (2013) introduced the so called "diversity parameters" to describe the particle mixing state. These parameters
are based on the calculation of Shannon information entropy which is often used to quantify uncertainties, and is different from the thermodynamic entropy used in other areas (Shannon, 2001). The calculation process is described in Riemer and West (2013), and the necessary calculation process for the CPMA-SP2 results is described in the supplements. In this study, each rBC-containing particle is considered to contain two types of material: rBC material and non-rBC material. The mass fractions of rBC core and non-rBC coating for both the single rBC-containing particle level and for the bulk rBC-containing particle
level that used for calculation are simply derived from the measured mass parameters. Briefly, the mixing state index ($\chi$) term can be given from:

$$\chi = \frac{D_\alpha - 1}{D_\gamma - 1} \qquad (3)$$

Where $D_\alpha$ and $D_\gamma$ represent the single-particle diversity and the bulk population diversity respectively and are used to describe the effective number of species in the single particle and the population of particles. The mixing state index ($\chi$) enables the precise quantification of the rBC-containing particle mixing states. For a well-mixed situation (internal mixing), $D_\gamma = D_\alpha = 2$, and the mixing state index $\chi = 100\,\%$. For an external mixing situation, $D_i = D_\alpha = 1$, and the mixing state index $\chi = 0$. In this study, because only the rBC-containing particles are detected, it is not possible to detect a completely external mixture.

### 3.3 CPMA-SP2 two variable distribution extrapolation

As introduced in the previous section, the CPMA-SP2 system measurement range set for this measurement campaign is up to $m_p = 15$ fg, which is roughly between 0.25 µm and 0.28 µm in equivalent volume diameter for typical urban rBC-containing particles, based on studies in other cities. However, during certain periods throughout the winter campaign, the distribution



unexpectedly extended to larger sizes outside of this range, meaning the overall rBC population was not completely captured by the measurement data. Some other studies regarding to the rBC-containing particle size distribution in China (Gong et al., 2016;Wu et al., 2017;Zhang et al., 2018b) show that the volume diameter of rBC-containing particles is larger than the largest CPMA selected size set in this study. To better quantify the bulk rBC mixing states and overcome the measurement range

limitation, the number concentration of rBC-containing particles is extrapolated beyond the measurement range. From the previous CPMA-SP2 results by Liu et al. (2017), the number of distribution can be assumed as log-normal. Hence, the inversion results are extrapolated up to $m_{\mathrm{p}} = 300$ fg, which is about 0.8 µm in equivalent volume diameter, by fitting the measured results with the simulated distribution function. The simulated distribution function is generated from the superposition of two lognormal distributions. Further details of the extrapolation process are presented in the supplementary material. Briefly, the

number distribution for $m_{\mathrm{p}}$ $(\frac{\mathrm{d}N}{\mathrm{dlog}m_{\mathrm{p}}})$ is extrapolated by fitting the log-normal distribution function. Two methods were then

attempted to extrapolate the two-dimensional distribution $\frac{\partial^2 N}{\partial \log m_{\mathrm{p}} \partial \log m_{\mathrm{rBC}}}$, and critically compared with a view to determining the optimal method. One was based on fitting the two-variable distribution distribution as a function of $m_{\mathrm{rBC}}$ (Fit $m_{\mathrm{rBC}}$ Method), and other based on fitting the two-variable distribution as a function of the mass ratio between the $m_{\mathrm{p}}$ and $m_{\mathrm{rBC}}$ (Fit Ratio Method):

$$\frac{\partial^2 N}{\partial \log m_{\mathrm{p}} \partial \log m_{\mathrm{rBC}}}\bigg|_{\text{Fit mrBC},(i,j)} = R_i \cdot \left( a_1 \cdot \exp\left( -\left( \frac{\log (m_{\mathrm{rBC},j}) - b_1}{c_1} \right) \right) + a_2 \cdot \exp\left( -\left( \frac{\log (m_{\mathrm{rBC},j}) - b_2}{c_2} \right) \right) \right) \quad (4)$$

$$\frac{\partial^2 N}{\partial \log m_{\mathrm{p}} \partial \log m_{\mathrm{rBC}}}\bigg|_{\text{Fit Ratio},(i,j)} = R_i \cdot \left( a_1 \cdot \exp\left( -\left( \frac{\log (m_{\mathrm{p},i}/m_{\mathrm{rBC},j}) - b_1}{c_1} \right) \right) + a_2 \cdot \exp\left( -\left( \frac{\log (m_{\mathrm{p},i}/m_{\mathrm{rBC},j}) - b_2}{c_2} \right) \right) \right) \quad (5)$$

While $\frac{\partial^2 N}{\partial \log m_{\mathrm{p}} \partial \log m_{\mathrm{rBC}}}\bigg|_{(i,j)}$ represents the number concentration of two variable distribution at $m_{\mathrm{p}}$ bin $i$ and $m_{\mathrm{rBC}}$ bin $j$; $R_i$ is

the ratio between the rBC-containing particles number concentration at the extrapolated $m_{\mathrm{p}}$ bin $i$ and the number concentration at the measured largest $m_{\mathrm{p}}$ bin (the bin with $m_{\mathrm{p}} = 15$ fg), $a_1$, $b_1$, $c_1$, $a_2$, $b_2$, $c_2$ are all the constants for the distribution function.

To validate the reliability of the extrapolation, the total rBC mass concentration from the CPMA-SP2 extrapolations are

compared with the SP2 data obtained during polydisperse sampling (shown in Figure 2). For the polluted periods where large rBC-containing particles exist, the Fit Ratio method may overestimate the extrapolated rBC core size, and as a result, the extrapolated rBC mass is significantly higher than the SP2 only results for the polluted periods. The comparison of Fit Ratio and Fit $m_{\mathrm{rBC}}$ extrapolation methods shown in Figure 2(a) illustrates that the Fit $m_{\mathrm{rBC}}$ method is closer to the rBC mass



concentration measured by the SP2 only method. For the summer results shown in Figure 2(b), the extrapolation results for the two methods are similar because the rBC-containing particles were smaller, and the extrapolation was less necessary.

In general, the Fit $m_{\mathrm{rBC}}$ method is better to appropriate the total rBC mass compared to the Fit Ratio method. Therefore, the
results in this study are extrapolated by the Fit $m_{\mathrm{rBC}}$ method.

## 4 Results and discussion

### 4.1 APHH campaign overview and CPMA-SP2 two-variable distributions

Figure 3 presents an overview of rBC and NR-PM$_1$ mass concentration for the whole winter and summer campaign. The rBC
and NR-PM$_1$ mass concentration varied significantly in winter, the rBC mass concentration increased to beyond $10\,\mu\mathrm{g}\cdot\mathrm{m}^{-3}$ while the NR-PM$_1$ mass concentration increased above $300\,\mu\mathrm{g}\cdot\mathrm{m}^{-3}$ during the polluted periods. In contrast, the rBC and NR-PM$_1$ mass concentration dropped down close to 0 during the clean periods. Comparing to the winter periods, the rBC mass concentration varied less significantly in summer. The rBC mass concentration varied between around 0 and $4\,\mu\mathrm{g}\cdot\mathrm{m}^{-3}$ while the NR-PM$_1$ mass concentration varied between 0 and $100\,\mu\mathrm{g}\cdot\mathrm{m}^{-3}$.

Figure 4 shows the two-dimensional distribution $\frac{\partial^2 N}{\partial\log m_{\mathrm{p}}\,\partial\log m_{\mathrm{rBC}}}$ graph which presents the measured BC mixing state. For better quantification, the measured CPMA-SP2 inversion have been 3-hour averaged. According to the measurement results from winter campaign period, three types of mixing state distribution under three different pollution levels were observed and these three types of pollutions are: Heavy Pollution (NR-PM$_1$ greater than $200\,\mu\mathrm{g}\cdot\mathrm{m}^{-3}$), Moderate Pollution (NR-PM$_1$
between 100 and $200\,\mu\mathrm{g}\cdot\mathrm{m}^{-3}$), and Light Pollution (NR-PM$_1$ less than $100\,\mu\mathrm{g}\cdot\mathrm{m}^{-3}$). These representative distribution under different pollution levels are shown in Figure 4(a–c). Figure 4(a) shows high concentrations of rBC-containing particles with large rBC cores and thick coatings were observed under heavy haze (fog) pollution conditions in the morning of 4th Dec. During that day a surface low pressure persisted in Beijing and its surrounding area, so air pollution accumulated to high concentrations. For the moderate pollution period shown in Figure 4(b), a large number of rBC-containing particles with
thinner coatings and smaller rBC cores were observed. The period of light pollution shown in Figure 4(c) has the smallest number concentration of rBC-containing particles among all the three pollution types but the shape of the two-variable distribution is similar to the moderate pollution period in that most of the rBC-containing particles were smaller than 10 fg and thickly coated BC particles were rare. As no highly polluted events comparable to winter were observed in summer, the $\frac{\partial^2 N}{\partial\log m_{\mathrm{p}}\,\partial\log m_{\mathrm{rBC}}}$ distribution in summer does not show as much variation during the whole campaign period - one example is





shown in the Figure 4(d). Initially, it would appear that the $\frac{\partial^2 N}{\partial \log m_\mathrm{p} \partial \log m_\mathrm{rBC}}$ distribution in summer is similar to the moderate or light pollution distribution in winter.

The one variable size distribution results from the integration of $\frac{\partial^2 N}{\partial \log m_\mathrm{p} \partial \log m_\mathrm{rBC}}$ is displayed in Figure 5. The volume

equivalent diameter for rBC-containing particles ($D_\mathrm{p}$) and the rBC core ($D_\mathrm{c}$) is also calculated by assuming a density of 1.2 g/cm$^3$ for rBC-containing particles and a density of 1.8 g/cm$^3$ for rBC cores for better comparison with other studies based on aerodynamic diameter. For the $m_\mathrm{p}$ number concentration size distribution ($\frac{\mathrm{d}N}{\mathrm{d}\log m_\mathrm{p}}$) shown in Figure 5(a), the mode particle size is between 2 fg and 10 fg for the light pollution and moderate pollution periods, while for the heavy pollution period there is a large increase in the number concentration for rBC-containing particles larger than 5 fg. The number concentration size

distribution for the $m_\mathrm{rBC}$ ($\frac{\mathrm{d}N}{\mathrm{d}\log m_\mathrm{rBC}}$) shown in Figure 5(b) also indicates that the rBC core is larger in the heavy pollution period in winter.

### 4.3 Coating thickness results

The coating thickness information is presented through the mass ratio (MR) parameter which is derived from the CPMA-SP2

inversion results. The MR parameter defined here follows the definition of Liu et al. (2017), which is given by:

$$\text{Single Particle MR} = \frac{m_\mathrm{non-BC,i}}{m_\mathrm{rBC,i}} = \frac{m_\mathrm{p,i} - m_\mathrm{rBC,i}}{m_\mathrm{rBC,i}} \qquad (6)$$

Where Single Particle MR is the mass ratio for rBC-containing particle $i$, and $m_\mathrm{non-rBC}$ is the non-BC material coated on rBC-

containing particle $i$, $m_{p,i}$ and $m_{rBC,i}$ are the mass of the rBC-containing particle $i$ and the mass of the rBC core for rBC-containing particle $i$ respectively. The traditional SP2 only method calculates MR by applying measured optical properties through the leading edge only (LEO) technique, and assumptions regarding the refractive indices and morphology are needed to determine the bulk relative coating thickness (Gao et al., 2007; Liu et al., 2014; Broda et al., 2018). The CPMA-SP2 method, however, is able to quantify the mass parameters more directly and without LEO fitting or a scattering inversion, therefore the

MR derived can be considered much more accurate. The bulk MR measured by CPMA-SP2 is calculated from:

$$\text{Bulk MR} = \frac{\sum_i N_i \cdot m_\mathrm{p,i}}{\sum_i N_i \cdot m_\mathrm{rBC,i}} - 1 \qquad (7)$$



Where $N_i$ is the number concentration for rBC-containing particle $i$. The comparison between bulk MR from CPMA-SP2 method and SP2 only method is shown in Figure 6. The bulk MR from CPMA-SP2 method present good correlation with the SP2 only method during the summer period and the winter period. Despite slight lower bulk MR observed from the SP2 only method, the bulk MR results from the two methods are generally close.

The CPMA-SP2 bulk MR time series results for winter and summer are presented in Figure 7. A significant increase can be observed during the winter haze period, and the bulk MR value reached a peak of 10 on December 4th. As indicated by the two-variable distributions during this haze period in the previous section, due to the presence of large numbers of large rBC-containing particles with their $m_{\mathrm{rBC}}$ between 1 fg and 10 fg, the bulk MR value increased rapidly. Influenced by micro-physical

process i.e. more active condensation and coagulation during the haze period, the non-rBC species were coated more heavily. Hence a large fraction of aged and thickly coated rBC-containing particles were formed caused by the rapid aging process during the haze period (Peng et al., 2016). In contrast to the heavy polluted periods, during the light polluted periods the rBC cores were thinly coated, which may indicate the particles were freshly emitted. The summer bulk MR results shown in Figure 7 illustrate that the bulk MR in summer varied between 1 and 2.7 which indicates the coating thickness was thinner in summer.

Though several moderate bulk MR values were observed, the bulk MR was generally small in summer.

The bulk MR frequency distribution is presented in Figure 8. Most of the rBC-containing particles were thinly coated with the bulk MR less than 2 in both winter and summer. The range of bulk MR values was also much broader in winter compared to the summer period. Numbers of thickly coated rBC-containing particles (bulk MR > 3) were observed in winter while no

thickly coated particles were found during the summer period. In general, the bulk MR frequency distribution from the CPMA-SP2 and SP2 only methods are similar.

A CPMA-SP2 two-variable distribution is able to present detailed single rBC-containing particle coating thickness information. The Single Particle MR can be derived to quantify the coating thickness on each rBC-containing particle. Liu et al. (2017)

stated that rBC-containing particles with an MR lower than 1.5 are typical for traffic emissions while an MR greater than 3 is typical for biomass burning emissions. The rBC mass fraction under different single particle MR level classifications is presented in Figure 8. Comparing the summer results with the winter results, the fraction of uncoated (Single Particle MR=0) and moderate coated (1.5≤Single Particle MR<3) rBC-containing particles is similar. The mass fraction varied slightly around 20% for uncoated rBC-containing particles at both summer and winter campaign periods. The thick coated rBC-containing

particles (Single Particle MR≥3) accounted for a higher fraction in winter, especially for the heavy haze periods between the 3rd and 5th of December. This large fraction of thickly coated rBC-containing particles was expected due to the coal burning in winter.  For thinly coated particles (0<Single Particle MR<1.5), it accounted for large fractions (around 60%) for the clean periods in winter and most of the time in summer which indicates that most of the rBC-containing particles were from traffic emissions during these periods.




## 4.4 Entropy parameters and rBC mixing state

Derived from the CPMA-SP2 inversions, the bulk diversity ($D_\gamma$) and the mixing state index ($\chi$) results shown in Figure 10 illustrate the variation of BC mixing state for the whole APHH campaign period. As the CPMA-SP2 system only detects the

rBC-containing particles and the number of species set here is 2, i.e. rBC and non-rBC material. To reiterate, this metric should be taken as the level of homogeneity of rBC mixing amongst the rBC-containing particles.

$\chi$ varied around 62% between 10th Nov and 21st Nov. A decrease of $\chi$ was observed between 19th and 20th Dec which indicates a period of more externally mixed rBC. Apart from some minor $\chi$ spikes, the mixing state of rBC-containing particles remained

unchanged between 10th Nov and 21st Nov. Whereas for the results between 1st Dec and 6th Dec, $\chi$ and $D_\gamma$ varied dramatically. After the polluted period on 30th Nov, a clean period occurred on 1st Dec and $\chi$ decreased rapidly to below 60%. When the haze formed between 2nd Dec and 4th Dec, $\chi$ increased gradually. On contrast, $D_\gamma$ decreased significantly during this heavy polluted period which was mainly caused by the thick coating material on the rBC cores leading to a reduction of rBC material mass fraction percentage. The haze subsided after $\chi$ reaches the peak of around 70% on 4th Dec, another clean period appeared

and $\chi$ and $D_\gamma$ resumes to the value before the haze. For the moderate polluted period between 6th and 7th Dec, $\chi$ reached another peak of around 68%, and the increase of bulk MR led $D_\gamma$ to decrease to around 1.5.

The mixing state of rBC-containing particles can be also associated with the air mass sources, and the relationship between the major air mass source, MR and rBC mixing state is further explored further in Figure 11. The winter characterisation result

is shown in Figure 10(a) and Figure 11(a). Among all the air mass sources, Plateau South and Western NCP were considered to represent the polluted air masses from which aged thick coated rBC-containing particles were transported during this period. When higher $\chi$ (between 60% and 75%) and bulk MR values (between 2 and 12) were observed the Plateau South air mass periods dominated. When Local air mass and Plateau North air mass accounted for more fractions, the presences of thin coated rBC-containing particles contributed to the significant increase of $D_\gamma$ values and rapid decrease of $\chi$ and bulk MR values. For

example, the reduction of rBC-containing particles internal mixture on 30th Nov and 5th Dec can be associated with a clean Plateau North air mass. This evolution of $\chi$ and $D_\gamma$ during the winter period agrees with the findings from Paris of Healy et al. (2014), who found that there is more internal mixing when polluted air masses dominate, compared to the local and clean marine air mass periods.

For the summer period shown in Figure 10(b), $\chi$ varied between 60% and 75%. As the bulk MR was lower in summer, the mass fraction of rBC material and the mass fraction of non-rBC material was closer within the bulk population, hence a slightly higher $D_\gamma$ was observed in summer. Though the bulk MR and the rBC mass concentration was not as high as that in winter,



sharp decreases in $D_\gamma$ together with the increase of $\chi$ still occurred occasionally in summer which indicate that some rBC-containing particles with moderate coatings were observed. Unlike the winter campaign period, the back-trajectory results in summer shown in Figure 11(b) do not show a significant association between mixing state parameters and air mass origin.

**4.6 Discussions**

In order to investigate which other parameters may act as a predictor for mixing state, Figure 12 shows the variation of $\chi$ in both summer and winter against NR-PM$_1$, bulk MR and rBC mass concentration. There was a strong correlation between the variation of $\chi$ and the variation of MR in winter as rBC-containing particles with thicker coatings tended to exhibit a more uniform internal mixture during the winter campaign period. For the summer period, however, no such significant correlation

was found. This observation indicates that higher bulk MR can be related to increased levels of internal mixing in winter, while the bulk MR metric cannot be used to predict the rBC mixing state in summer. Although the rBC-containing particles were not thickly coated in summer, and the pollution level was lower, the rBC-containing particles were still well mixed.

Comparing between two seasons, there was more diversity of internal mixing state of BC in summer than in winter under the

similar bulk MR level condition or similar NR-PM$_1$ and rBC mass concentration level. About 30% of the experimental period in summer had $\chi$ over 70% in summer, however this fraction was almost absent in winter. This means although the rBC mass loading was low in summer however a more homogenous distribution of coatings or internal mixing state was present. The rBC-containing particles in winter, however, showed lower internal mixing, which may be in line with the source apportionment study that about 64% of rBC-containing particles were from primary sources in winter experimental period

(Wang et al., 2019; Liu et al., 2018a). These primary sources of rBC-containing particles may have coatings externally mixed with the rBC cores, leading to lower $\chi$ shown here. Due to the higher pollution level, i.e. more pre-existing particles, the non-rBC materials could be coated on rBC cores through coagulation process, which may result in the higher relative coating amount in winter. An increasing trend of $\chi$ was shown when increasing the bulk MR and NR-PM$_1$ level, and this means the higher pollution will cause both enhanced coating content and more homogenous distribution of coatings on rBC cores.

In summer, the primary emissions are less compared to winter, and SOA contributes more in Beijing (Sun et al., 2018; Hu et al., 2016). Previous studies show that an internal mixture would be expected for BC associated with a larger secondary fraction (Krasowsky et al., 2018; Bondy et al., 2018). In addition, the rBC-containing particles experience more intense photochemical processing in summer, which may promote the formation of secondary species internally mixed with BC (Xu et al., 2018).

However, this does not mean the amount of coatings on BC will be necessarily higher in summer because the higher temperatures and enhanced dilution may cause the primary semi-volatiles to favour the gas phase.





The results here suggest that the increase of coating content such as the bulk MR above 4 could importantly increase the internal mixing state of rBC-containing particles, as indicated by the winter results; but the mixing state of rBC-containing particles should also depend on the coating formation mechanism, such as in summer still showed a higher homogenous mixing at low pollution level. Both primary source influence and secondary coating formation mechanism should be considered in interpreting the BC mixing state in the atmosphere.

Before comparing with previous studies, it should be noted that the definition of internal mixture defined here may differ from some other studies. Following the definition in Riemer and West (2013), the "fully internal mixture" defined in this study refers to the bulk particles consist of the same species, and the species account for the same amount percentage within each particle ($D_\gamma = D_\alpha$).

## 5 Implications

The CPMA-SP2 together with a new inversion method is capable of exploring BC mixing state in a way not previously possible and results can be directly compared with particle-resolving models such as PartMC-MOSAIC (Riemer et al., 2009) when simulating the behaviour of BC in polluted plumes. The fact that the MR parameter is able to predict the mixing state variation under certain conditions but not others may have implications for detailed, non-explicit models like GLOMAP (Mann et al., 2010). It is known that being able to predict the transition of BC particles from fresh ('hydrophobic') to aged ('hydrophilic') is a major sensitivity in climate models (Koch et al., 2009). The assumption complete internal mixing in bulk models may lead to an overestimation in BC absorption and hygroscopicity, further the CCN activities may be overestimated as well (Fierce et al., 2017). Some models consider this ageing of BC particles through coating and the resulting changes in CCN and optical properties, for example GLOMAP (Mann et al., 2010). However, our results suggest that it may be appropriate to assume internal mixing of BC in summer, though the influence on optical properties may not be as important as in winter due to thinner coatings. These internally mixed rBC-containing particles may serve as CCN (Oshima et al., 2009) and exert impacts on cloud microphysics. Some model studies have also pointed out that large uncertainties in the scavenging efficiency of BC in model studies remain because of the underestimation of BC life time (e.g. (Myhre and Samset, 2015)), and these uncertainties arise partly from the lack of BC coating thickness and size distribution observation (Taylor et al., 2014). In addition, there is a significant overestimation of BC absorption enhancement for the model studies that assume the BC mixing state as population-averaged (Fierce et al., 2016). More detailed coating information (MR results) and mixing state information presented in our study can contribute to the simulation of the BC lifetime and transportation in the future. In general, more appropriate mixing state treatment of BC particles in models can reduce the uncertainty in BC lifetime and its role as CCN or ice nucleation (Bond et al., 2013; Fierce et al., 2017).



## 6 Conclusions

In this study, the combination of a centrifugal particle mass analyser (CPMA) and a single particle soot photometer (SP2), has been applied to the investigation of mixing state of refractory Black Carbon (rBC)-containing particles in urban Beijing. A novel CPMA-SP2 inversion algorithm is able to present the detailed two-variable number distribution as a function of rBC

mass and total rBC-containing particle mass. As the CPMA-SP2 system directly measures the particle mass parameter, more accurate mass ratio (MR) results for the rBC-containing particles are reported. With the detailed inversion metric results, the mixing state index ($\chi$) is derived to quantify the mixing state of rBC-containing particles. The measurement results show that the measured bulk MR during the winter campaign period varied significantly from around 2 to 12, while the $\chi$ value varied between 55% and 70%. The highest bulk MR values were associated with the haze polluted periods in winter, and the $\chi$ value

reached higher values under these conditions (over 65%), which illustrated there was more internal mixing during this period. This result shows that higher pollution level will cause both enhanced coating content and more homogenous distribution of coatings on rBC cores in winter and makes the bulk MR may act as a predictor of mixing state. Meanwhile, the back-trajectory analysis indicates that polluted air from Southern Plateau dominated the aged rBC-containing particles in Beijing during winter. For the summer campaign period, the bulk MR varied between 2 and 3. The $\chi$ varied between 60% and 75%, but was

found to be independent of MR. The slightly higher $\chi$ in summer indicates that internal mixing is preferred for rBC particles which may be caused by higher secondary concentrations. However, thick non-rBC coatings were not favoured in summer due to the limited polluted events and high temperature in summer, and further no apparent correlation relationship was found between the rBC-containing particle mixing state and air mass sources. The observations presented here have implications for detailed models of BC, its optical properties and its atmospheric lifecycle, necessary to assess its local and global impacts. In

models that consider mixing state, it is often either assumed or related to its MR. The methods and results discussed here can better describe the rBC mixing state and coating thickness which may offer new insights and contribute to improvements in the accuracy of simulations.

**Data availability**

Processed SP2 data is available through the APHH project archive at the Centre for Environmental Data Analysis (http://data.ceda.ac.uk/badc/aphh/data/beijing/). Raw data is archived at the University of Manchester and is available on request.

**Contribution**

D.L. and J.D.A. designed the research; C.Y., D.L., J.D.A., R.J., H.C., Y.S. and P.F. performed experiments; K.B. and J.O. provided the initial inversion code; C.Y. improved the inversion code and performed the CPMA-SP2 data analysis; D.L.



performed the initial SP2 data processing and HYSPLIT runs; Y.S. analysed the NR-PM$_1$ data; C.Y., D.L. and J.D.A. wrote the paper.

## Acknowledgments

This work was supported through the UK Natural Environment Research Council (grant refs. NE/N007123/1, NE/N00695X/1) and the National Natural Science Foundation of China (grant nos. 41571130024, 41571130034 and 21777073).

*Competing interests*: The authors declare that they have no conflict of interest

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

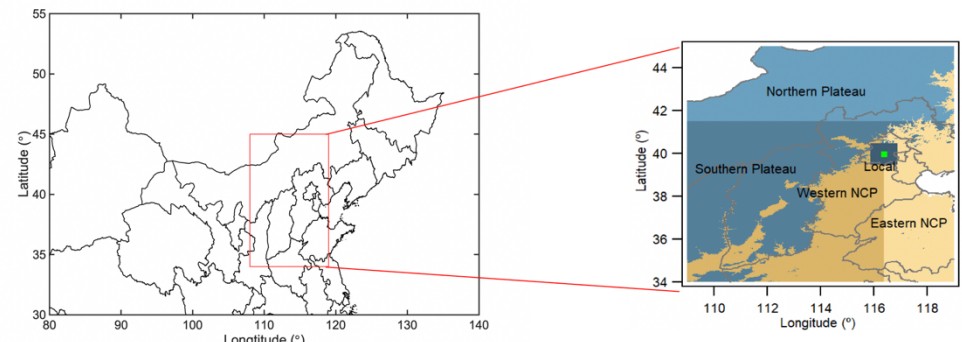

Figure 1 Classification of air mass sources (modified from Liu et al. (2018a))

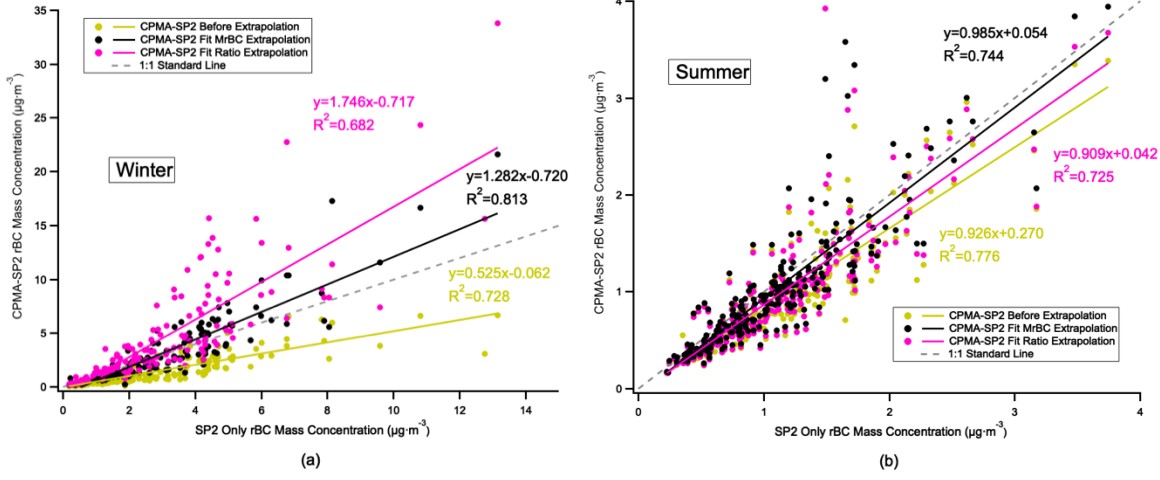

Figure 2 The rBC mass concentration comparison between SP2 only measurement and two different CPMA-SP2

extrapolation methods in winter (a) and summer (b)



Figure 3 Time-series of rBC mass concentration and NR-PM$_1$ concentration for the winter (a) and summer (b) campaign

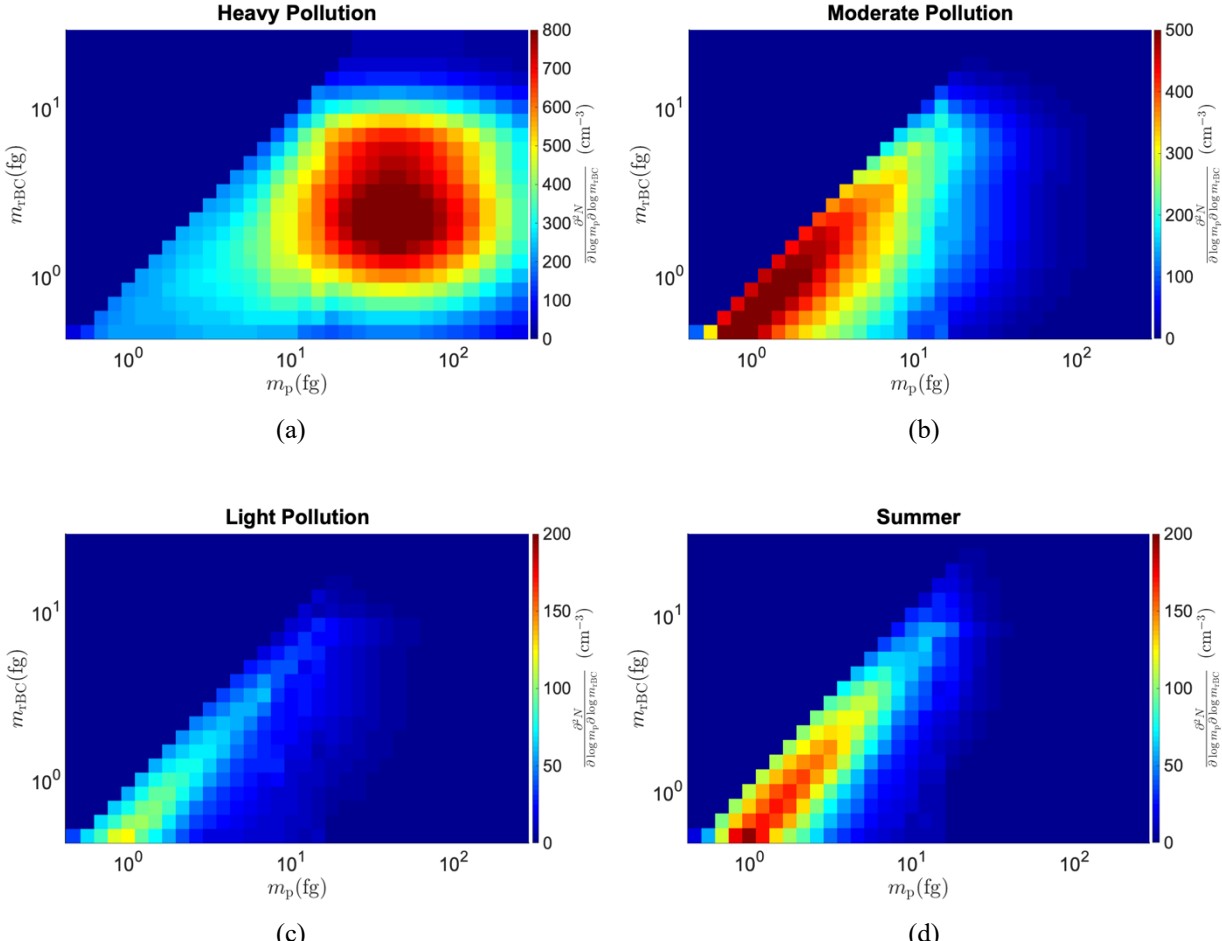

(a)            (b)

(c)            (d)

Figure 4 CPMA-SP2 Inversion matrix examples under different pollution conditions in winter (a, b, c), and the example in the summer period (d), the $x$-axis, $m_p$, represents the mass of the whole rBC-containing particle while the $y$-axis, $m_{rBC}$, represents the mass of the rBC core. The colour bar represents the magnitude of the two-variable distribution with a number concentration unit as $cm^{-3}$. All the bins with $m_{rBC}$ larger than $m_p$ are zero since such a particle does not exist in reality, and therefore the two-dimensional function distribution graph has a shape of triangle.





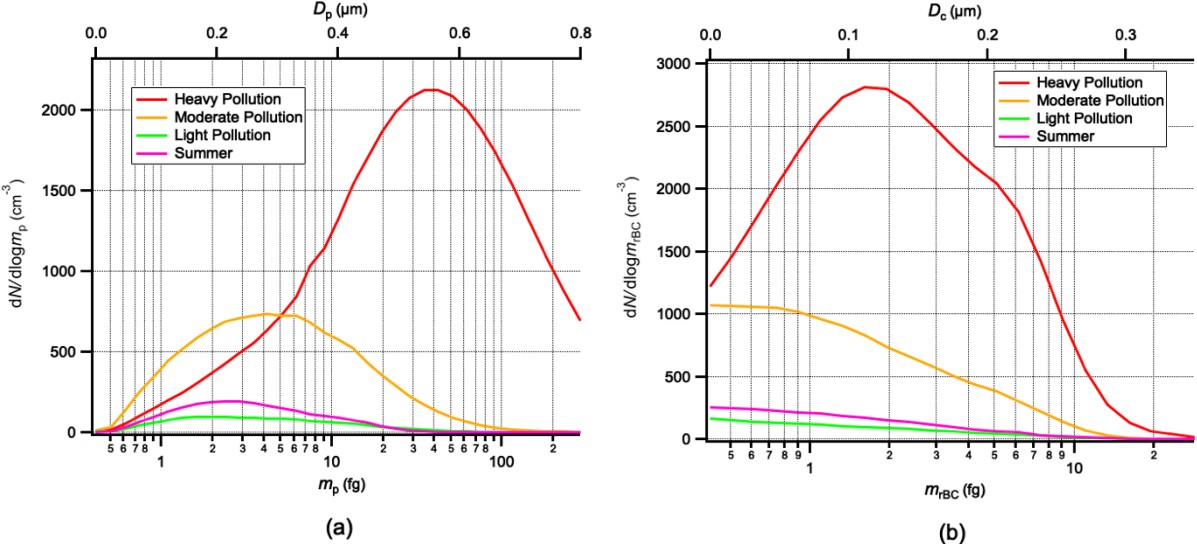

Figure 5 CPMA-SP2 $m_\mathrm{p}$ number distribution (a) and $m_\mathrm{rBC}$ number distribution (b) under different pollution conditions

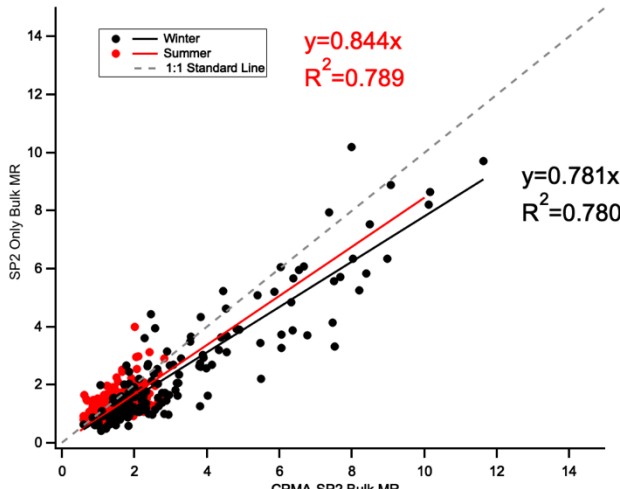

5         Figure 6 The comparison between bulk mass ratio (MR) from CPMA-SP2 method and from SP2 only method



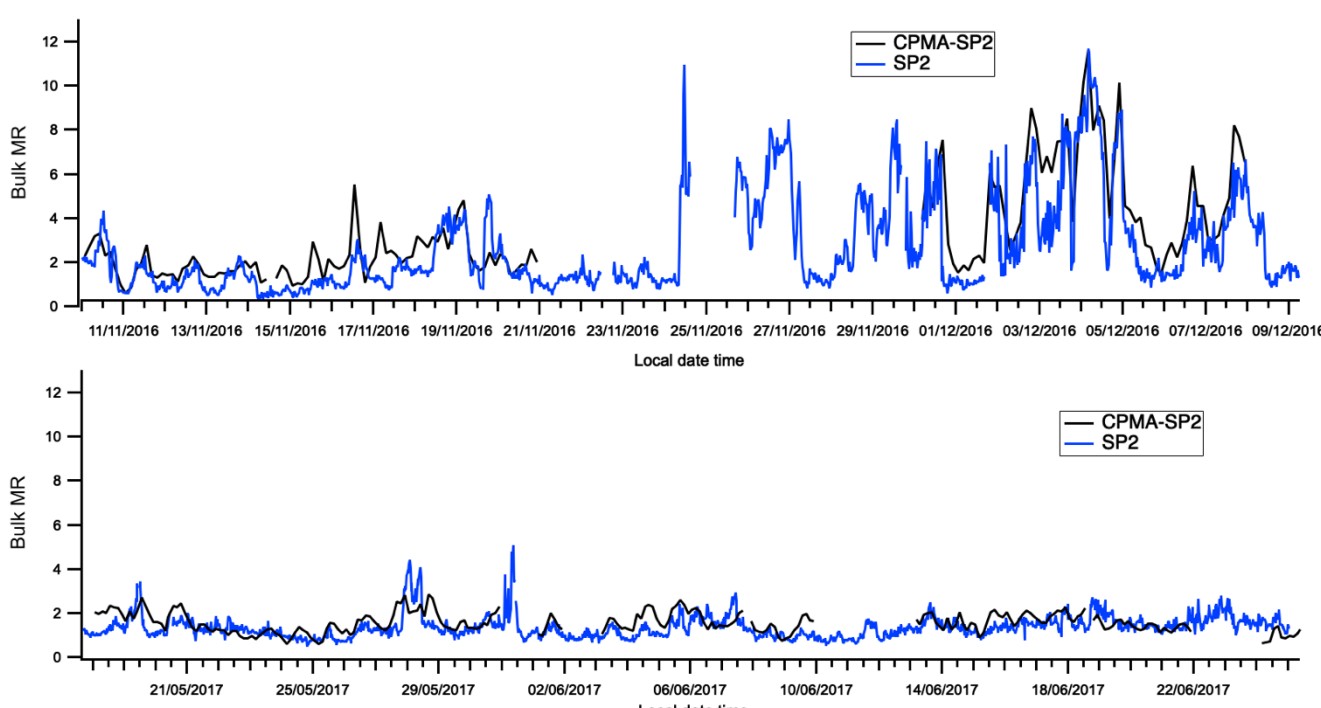

Figure 7 Time series of bulk MR for winter and summer

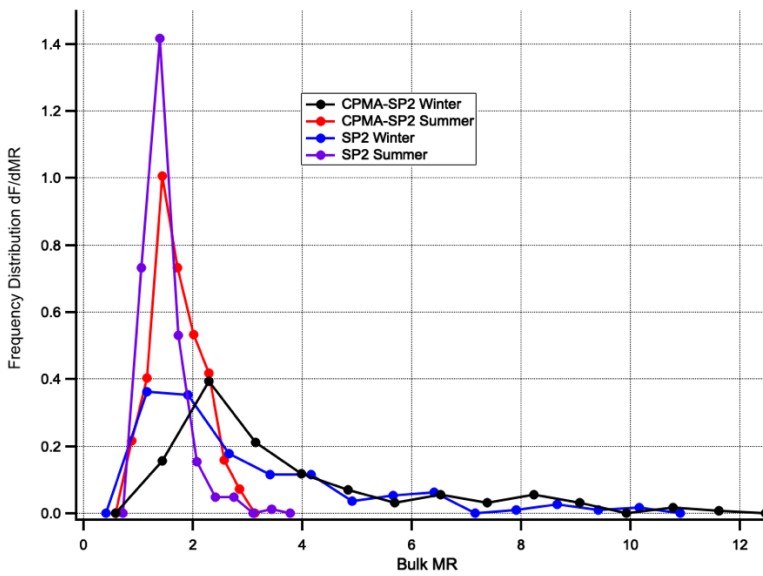

Figure 8 Frequency distribution of bulk MR from both CPMA-SP2 and SP2 only method





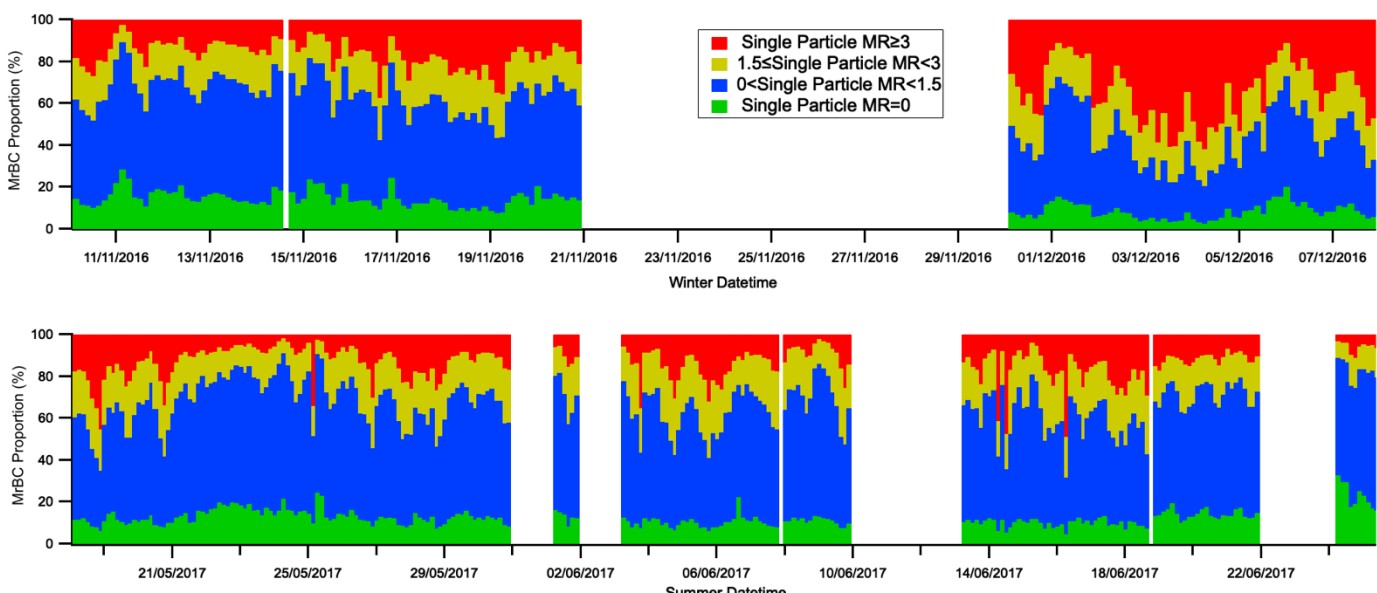

Figure 9 The revolution of $m_{\mathrm{rBC}}$ proportions at different MR level for winter and summer

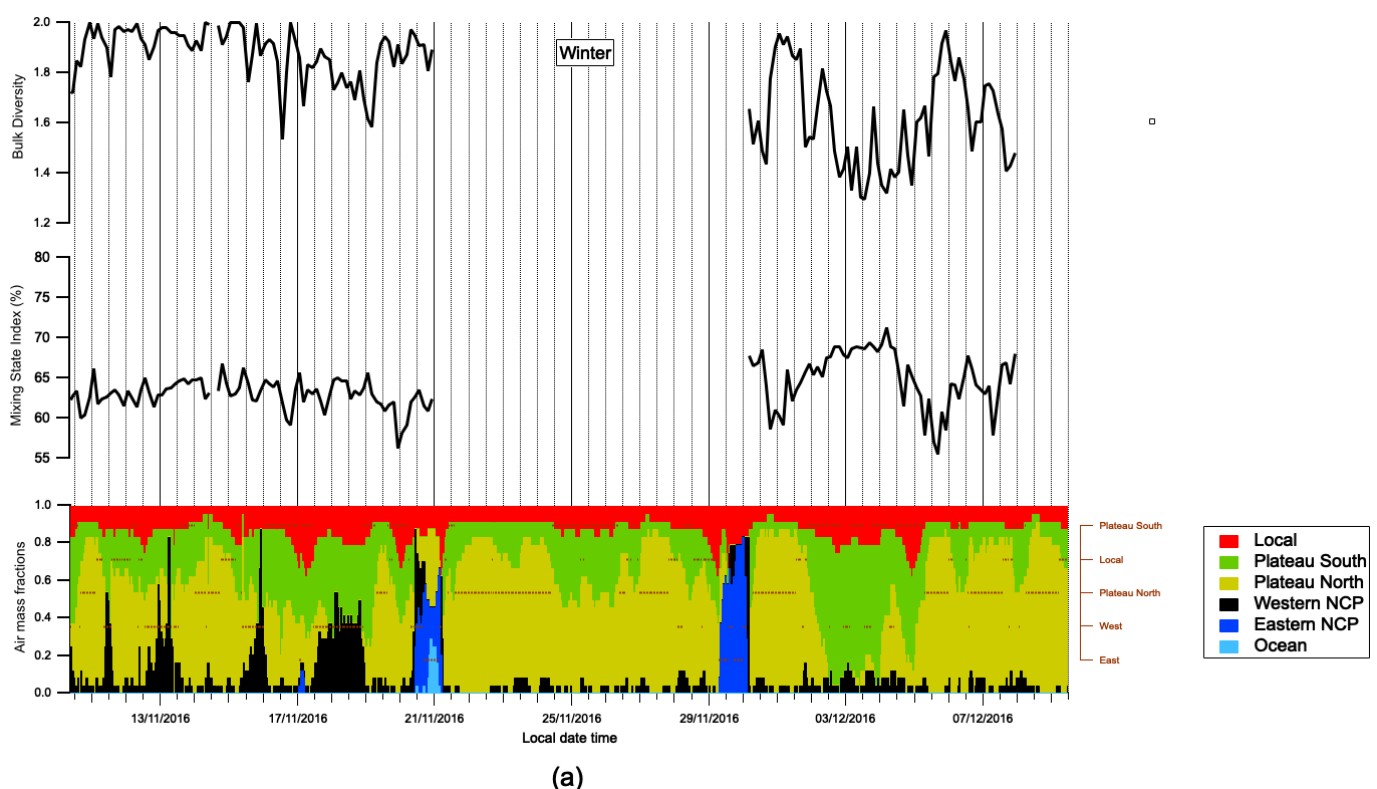

(a)

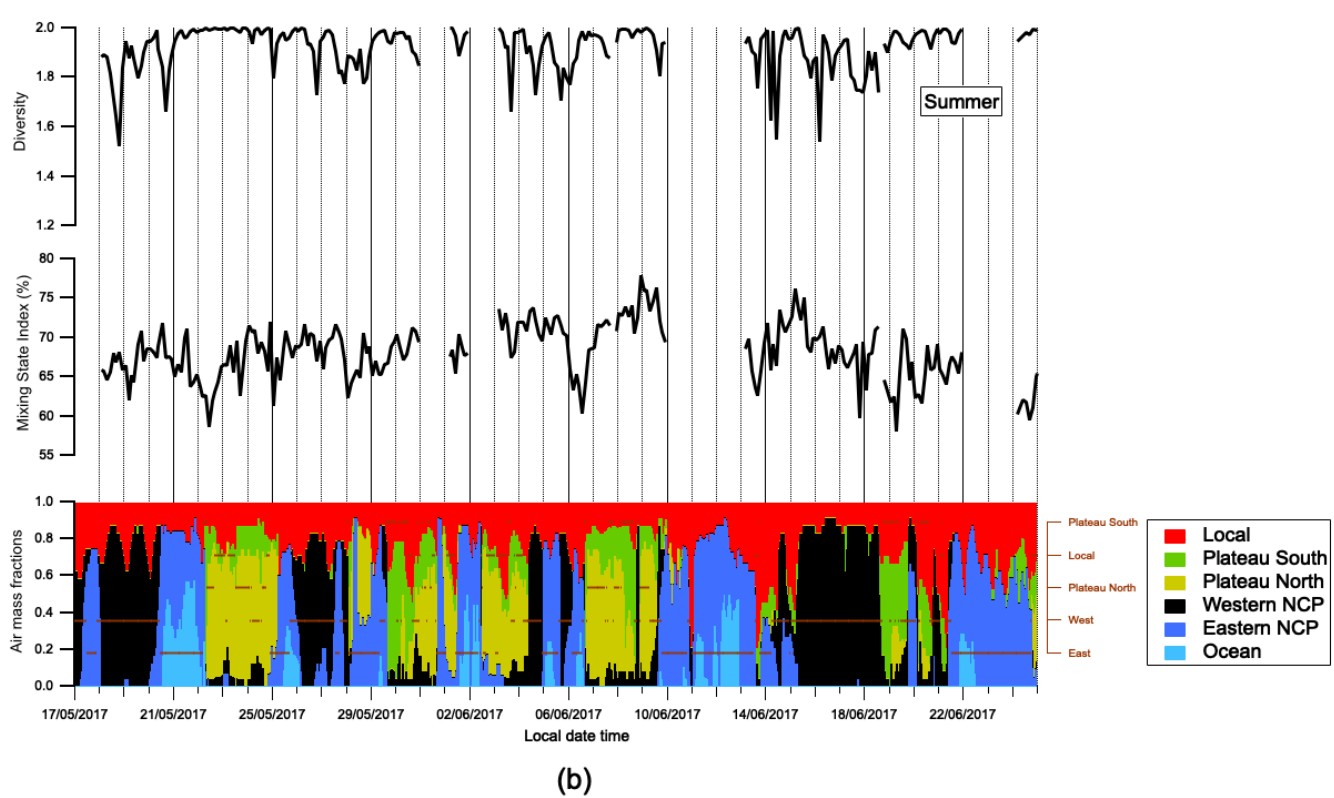

Figure 10 Bulk Diversity and Mixing State Index for the winter (a) and summer (b)

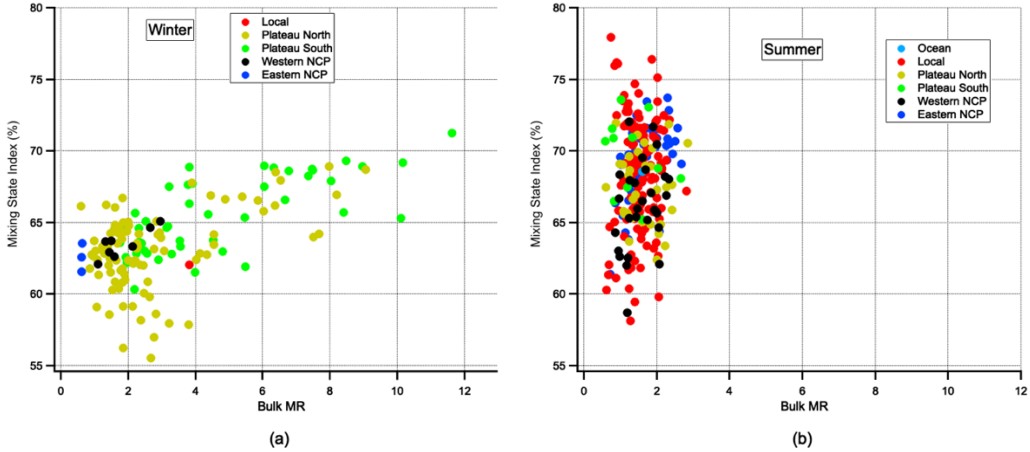

Figure 11 Air Mass Sources contribute to the MR and Mixing State Index for summer (a) and winter (b)



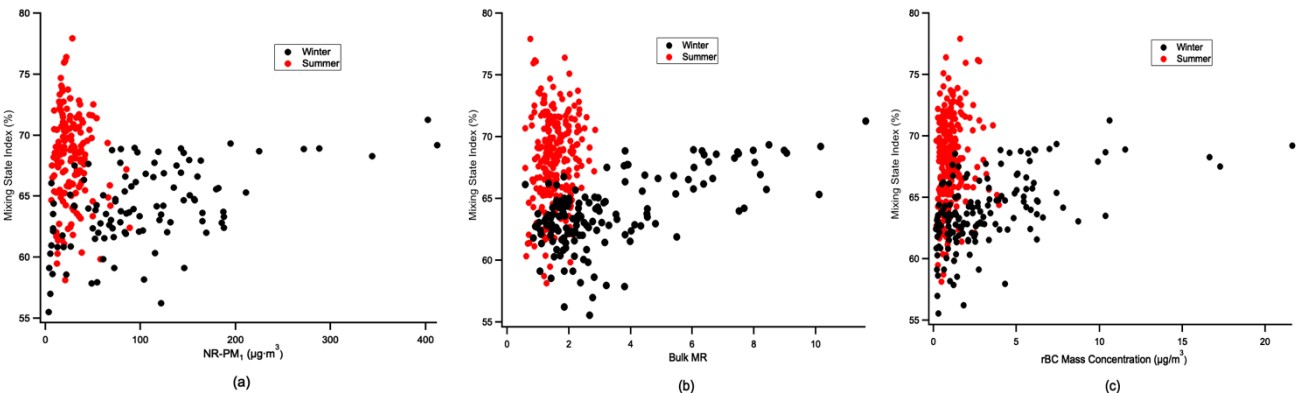

Figure 12 (a) The relationship between NR-PM$_1$ and Mixing State Index, (b) the relationship between Mass Ratio (MR) and Mixing State Index, and (c) the relationship between rBC mass concentration and Mixing State Index

