# Peer review of "Characterising mass-resolved mixing state of Black Carbon in Beijing using a morphology-independent measurement method"

_Atmospheric Chemistry and Physics, 2019_

## Referee Comment (RC1) · Anonymous Referee #1 · 20 Jul 2019

This paper presents results from the APHH campaign in Beijing where mixing state of black carbon-containing particles was experimentally determined using a combination of a centrifugal mass analyzer and a single particle soot photometer. To quantify "black carbon mixing state", 2D number distributions were produced, the mass ratio of non-BC and BC material was used, as well as the mixing state parameter chi.

This paper represents an important step forward in deriving mass-fraction-based metrics from single-particle measurements. I commend the authors for accomplishing this, as it is a key prerequisite for eventually being able to compare these measurements to mixing-state-aware models. The data analysis reveals interesting and novel insight

into the seasonable variability of black carbon mixing state in a polluted environment such as Beijing. Overall this work warrants publication in ACP, but before it can be accepted, several issues need to be addressed. I have four major comments regarding the presentation and interpretation of the results, and a number of minor comments, mainly pertaining to the clarity of notation and terminology.

**Major comments:**

1. The usage of the mixing state chi is confusing in the paper. Due to the measurement technique, it is only the mixing state of BC-containing particles that can be evaluated, i.e. the method is blind to a potentially existing subpopulation that only contains non-BC material. This is an important point, and it is explained in the paper, but it comes too late. I recommend introducing the specific use of chi already in the introduction, following the overview of previous studies that used chi as a metric for mixing state (note that each of those studies have their own take on chi). The authors might consider calling their chi "$\chi_{BC}$" or similar, to emphasize this point.

2. Related to comment 1, the interpretation of chi is also inconsistent throughout this paper. In the abstract and a few times in the text, it is correctly mentioned that chi (as defined here) "quantifies whether the coating is evenly distributed across the rBC-containing particle population". – I agree with this interpretation, and suggest that the authors explain this in the text more precisely, and stick to this interpretation throughout the text. I.e. a high value of chi means that all particles have the a very similar BC mass fractions (the same mass fraction for chi=100%) compared to a low value, where the distribution of BC mass fractions is very uneven across the population.

I strongly suggest not to use language like "A decrease of chi was observed [. . .] which indicates a period of more externally mixed rBC" (p. 11, line 8). The phrase "Externally mixed" is not meaningful here. In fact, I believe you can describe your results without even mentioning the terms internally and externally mixed.

3. Having established the meaning of chi in your study, it is important to recognize that

the diversity metrics as used here are mass-weighted. Therefore, the overall chi for the whole population will represent mainly the conditions for the large particles. It would be interesting to see how chi varies between small and large particles. Is it possible to calculate a size-resolved chi?

4. Use of the terminology "core-coating". The authors emphasize that their measurement method is morphology-independent, but still frequently use the terms "BC core" and "coating". I think these terms should be avoided because it invokes the traditional BC core-shell assumption, which may or may not apply in reality (and which is actually not made by the authors!).

**Minor comments:**

1. The abstract is too long. I recommend shortening it.

2. p. 1, line 17: Which year was the campaign in?

3. p. 1, line 25: "positively associated" - Do you mean "correlated"?

4. p. 2, line 9: Reference to Ramanathan and Carmichael (2008) is more than 10 years. Suggest to replace with a more recent reference.

5. p. 3, line 20: "measure the different mass ratio" – Should read "measure the mass ratio"?

6. p. 4, line 25: The size ranges for the detection limits are confusion. Do you mean 200-720 nm for the mixed particles (i.e. referring to the scattering signal) and 70-850 nm for the BC-component (traditionally called the BC core)? Please clarify. In this context, Figure 5 shows that the distributions extend to smaller diameters than 200 nm/70 nm. Do you also extrapolate towards the small sizes (not only the large sizes)?

7. p. 5, section 2.5: This is beyond the scope of this paper, but possibly an avenue for future research: Since you know the size distribution of the non-refractory particle matter from the AMS, and you know how much NR material is associated with BC,

can you infer the size distribution of the subpopulation that does not contain BC by combining the AMS with the CPMA-SP2 measurements? This would yield a complete picture of aerosol mixing state.

8. p. 6, line 22: This is an example where Major comment 2 applies. The terms internal/external mixing are not useful here. Chi as applied to the subpopulation of BC containing particles quantifies how evenly the BC and non-BC mass fractions are distributed. (with a focus on the large particles).

9. p. 8, line 16: "For better quantification" – what does that mean? To reduce noise?

10. p. 9, line 1: "It would appear that .. " – can you please clarify? Do you mean that the summer distribution is similar to winter/moderate or winter/light pollution, or that it is not similar?

11. p. 9: equation 6 and 7: Suggest finding more succinct variable names for "single particle MR" and "bulk MR". Even $MR_i$ and $MR_{bulk}$ would be an improvement here.

12. p. 9, equation 7: Since $m_{p,i}$ refers to an individual particle, and the summation is over all particles, there should not be the factor $N_i$. In other words, isn't $N_i = 1$?

13. p. 10, line 8: "due to the presence of large numbers of large rBC-particles … the bulk MR value increased rapidly." This seems the wrong way round, since $m_{BC,i}$ is in the denominator for Bulk MR.

14. p. 10, line 27: should be Figure 9. Is there anything that could be gained by comparing the bulk MR with the population average of the single particle MR?

15. p. 10: Figure 9: What does MrBC mean? Caption: Should read "evolution".

16. p. 10, line 32: Can you please add a reference, why coal burning should lead to thickly coated rBC-containing particles?

17. p. 11: Comparing to Healy et al. (2014), it would be important to note that they defined chi based on the 5 species system (SO4, NO3, NH4, OA, BC). So those chi

values and the one from this study are not directly comparable.

18. p. 11: Figure 10, top panel: Specify that it's bulk diversity for the y label.

19. p. 12, line 6: "there was a strong correlation. . ." Can you quantify this more precisely?

20. p. 12 Discussions: This section is also a good example that should be rephrased in light of Major comment 2. I think the main point of figure 12 is that during summer, for a narrow range of bulk MR a wide range of chi is observed, meaning even for similar bulk MR, there is a variation how the "coating material" is distributed (note that in summer chi is not always high). During winter, there seems to be a positive correlation between bulk MR and chi, meaning that the NR material is more evenly distributed when bulk MR is high. The explanation of why this summer/winter difference happens, however, is not clear. It would be important to improve this discussion.

21. p. 13: Implications: I appreciate the authors effort to connect their results to the modeling community, but The statement "it may be appropriate to assume internal mixing of BC in the summer" is over-reaching in my opinion. It would at least require an estimate of calculating absorption enhancement for the conditions shown here to substantiate. Water uptake (which will be modulated by the composition of the NR material) further impacts this impact.

I think an important contribution of this study is that it provides a framework to produce the 2D distributions as shown in Figure 4 (mass-based, including an objective method of extrapolation). This is data that mixing-state aware models (including, but not limited to particle-resolved models) can compare to.

22. Axis labels are generally too small
* * *

---

## Referee Comment (RC2) · Anonymous Referee #3 · 9 Oct 2019

**General remarks:**

The introduction provides a good overview on reasons why a better understanding of the mixing state of atmospheric black carbon (BC) particles with other particulate matter is required when it comes to e.g. their climate impacts through aerosol-radiation and aerosol-cloud interactions. This prepares the ground for the single main topic of this study.

In a recent lab study, the combination of a coquette particle mass analyzer (CPMA) and a single particle soot photometer (SP2) was shown to be a useful and reliable means to quantify the 2-dimensional mixing state distribution as a function of total particle mass and BC core mass, which also works for complex BC aerosols with high mixing state diversity on a single particle level. Furthermore, such measurements were not accessible with comparable data coverage and accuracy to other previously applied experimental methods. The study here takes the combined CPMA-SP2 approach for the first time to the field, specifically to an air pollution hotspot where the environmental impacts of aerosols are of great interest. This certainly makes the resulting data set on BC mixing state and mixing state diversity on single particle level worth publishing.

Unfortunately, the manuscript was hastily prepared and not really at the level expected for a high quality scientific journal. Reasons for this judgment are given in below comments and include:

- Very imprecise and sometimes wrong statements concerning the mixing state index "chi", which is the very central topic of this study.
- Insufficient sensitivity analyses and discussion on how the results for the mixing state index "chi" may be affected by the detection limits of the applied instruments.
- Week discussion and interpretation of the results.
- Poor language at many instances, definitely below the level that can be expected from joint forces of the whole author list.

**Major comments:**

1. P6, L24: I am puzzled about the statement "In this study, because only the rBC-containing particles are detected, it is not possible to detect a completely external mixture." – I would expect that the formalism was spitting out chi=0 (indicating completely external mixture for the particle types under consideration), if the coupled CPMA-SP2 measurements would provide m_rBC = m_p for every single particle (or more precisely: when calculating the limes towards this mixing state as the equations contain multiplication of "zero times infinity" or division of "zero by zero" for a perfectly externally mixed population).

2. Truncation of the BC particle mixing state 2D-PDF due to detection limits of the applied instruments is often unavoidable, and so it is in this study. To my judgement, the authors have done a fair job in addressing the impact on the resulting mixing state parameter from the upper limit of measured total particle mass (at least for 3 out of the 4 examples; separate comment on the latter follows). However, they have not at all addressed the impact of the SP2 lower detection limit for rBC mass in a single particle. The 2D-PDFs very nicely illustrate that for total particle mass equal to the SP2 lower detection limit only externally mixed BC particles can be measured, while internally mixed BC with, consequently, smaller BC core remains undetected. From a pessimistic perspective, the nearly complete mixing state information for BC containing particles is only available for either a narrow range of BC core sizes (roughly indicated by red square), or for a narrow range of total particle mass (roughly indicated by the magenta square).

[Figure]

How does the overall mixing state diversity parameter compare with its value calculated for the above two constrained size ranges? Does this hamper absolute interpretation of the chi values or is this only a minor issue?

Personally, I do not really have a feeling for the sensitivity of chi to such truncation effects. However, the fact that chi appears to be weighted by particle number and that particle number is dominated by small particles in the range of the lower SP2 cut-off would suggest that truncation at the lower SP2 cut-off could have a much more dramatic effect on the resulting chi value than the truncation at the upper CPMA cut-off in many cases (the heavy pollution period is obviously the counter-example).

3. Taking the previous comment a step further would raise the question whether a number-weighted parameter is suitable when it comes to e.g. BC lensing effect, where we care in first order approximation about the BC core size range where the BC mass size distribution peaks. Would it also be useful to additionally compute the particle diversity diameter exclusively for a narrowish range of rBC core mass around the modal size of the rBC mass size distribution (which will require some extrapolation above the upper CPMA cut)? This might also provide a mixing state information that is more readily comparable across different studies with similar instruments as they only have to cover this range well, whereas the actual lower/upper instrument cut-offs would loose out in relevance.

4. The extrapolation of the mixing state PDF looks reasonable for the example shown in the SI (and likely for the low/medium pollution winter and the summer cases). However, showing the heavy pollution period mixing state results without indicating the measured/extrapolated parts of the 2D-PDFs is really misguiding the reader:

[Figure]

While the measured part still provides valid information on the fact the BC is generally highly internally mixed, the mixing state diversity parameter chi, which is vastly dominated by the extrapolated part of the 2D-PDF, can hardly be meaningful in absolute terms.

5. P10, L6-13: While not being wrong, the discussion in this paragraph is rather weak. In the end the observed mixing state is a result of various factors including mixing state at emission, average atmospheric residence time of the BC particles, representative aging rate (in the sense of coating acquisition rate), etc.

   a. For the high pollution winter case the authors simply say: high secondary aerosol production → more condensation → more rapid coating acquisition → higher MR. This is incomplete.

   b. For the low pollution winter case the authors simply say: observed thin coatings may indicate fresh emission. This is also incomplete.

   c. The statement "due to the presence of large numbers of large rBC containing particles with their $m_{rBC}$ between 1 fg and 10 fg, the bulk MR value increased rapidly" does not really make sense (BC core size alone cannot explain anything about MR).

   d. Further down (P10, L30) the following statement is made for the high pollution period: "This large fraction of thickly coated rBC-containing particles was expected due to the coal burning in winter." – How does this relate to above arguments and where the coal burning BC mixing state knowledge come from?

6. Presentation of results shown in Fig. 9 (last paragraph of Sect. 4.3): The potential evolution of MR during aging after emission has strong implications on which type of conclusions can or cannot be drawn. This is not properly acknowledged in the current discussion which is, therefore, flawed.

7. P12, L16: The following statement is made here: "This means although the rBC mass loading was low in summer however a more homogenous distribution of coatings or internal mixing state was present." – What is the expected process that causally links BC mass concentration with BC mixing state homogeneity? (Or do you have cross-correlations between BC mass concentration and other parameters that causally relate to BC mixing state diversity in mind?). The point raised in this comment also questions the meaningfulness of Fig. 12c.

**Minor comments:**

8. P1, L25 (abstract): The "or" in the following statement sounds confusing: "χ of rBC-containing particles was highly positively associated with increased bulk MR, rBC mass loading or pollution level in winter,…"

9. P1, L29 (abstract): The following statement is difficult to understand and hardly self-explaining without having read the manuscript: "The same level of bulk MR corresponded with a higher χ in summer than in winter and this tended to suggest a limited formation of coatings on rBC largely depended on primary sources."

10. P1, L32 (abstract): What is the mechanism that links ambient temperature with BC coating thickness?

11. P1, L34 (abstract): The statement "The mixing state of rBC-containing particles should also depend on the coating formation mechanism, both primary source influence and secondary coating formation mechanism should be considered in interpreting the rBCcontaining particles mixing state in the atmosphere." should rather be made early in the abstract, i.e. before providing interpretations of observed BC mixing state degree and variations.

12. P2, L18-21: the Shiraiwa 2007 study cited here in the context of relationship between BC mixing state and its droplet activation does not provide any activation measurements. Schroder et al. (2015) or Motos et al. (2019) are examples of recent studies which directly investigated the activation of BC particles in ambient clouds as a function of their size and mixing state.

13. Sect. 2.4 on CPMA-SP2 coupled system: Are small uncharged particles passing the CPMA in addition to the charged and mass selected particles an issue (upper cut-off size for uncharged particles depends on rotational speed)?

14. P8, L12-14: Measurements of major chemical components of atmospheric aerosols rarely provide "zero". Instead, concentrations can drop by several orders in magnitude or below the lower limit of quantification. Therefore it is not very useful to report observed value ranges as "from zero to ….".

15. Figure 3: The period marked as "light pollution" looks like what is described as "clean" in the first paragraph of Sect. 4.1. Actually, a few lines further down (P.8, L20) I came across the definition "light pollution means NR-PM1 < 100 ug/m3", which is not congruent with the shading shown in Fig. 3.

16. P9, L5: Here it says: "The volume equivalent diameter for rBC-containing particles ($D_p$) and the rBC core ($D_c$) is also calculated by assuming a density of 1.2 g/cm3 for rBC-containing particles and a density of 1.8 g/cm3 for rBC cores." – Strictly speaking, different densities should be assumed for core and coating then it comes to estimating $D_p$ from mp. Furthermore, an average density of 1.2 g/cm3, which is lower than that of BC, inorganic salts and likely a fair fraction of the organics, appears to be a bit low. However, doing it more precisely would likely have little effect on the resulting Dp values.

17. P9, L9-11: The shift of the rBC core size distribution to larger sizes appears to be quite dramatic (it would definitely look dramatic when shown as mass rather than number size distribution).
    a. Are these rBC core size distributions based on the polydisperse SP2 measurements or on the (extrapolation dominated) coupled CPMA-SP2 measurements? I would only trust the latter for the high pollution period (unless that one was jeopardized by coincidence issues).
    b. How do these findings on rBC core size distribution compare with existing literature (if any)?

18. Sect. 4.3: The claim is made here that the CPMA-SP2 combination more accurately provides the MR than SP2 only data (combining incandescence signals with LEO-fit based optical sizing). This claim seems fully justified. A quite interesting question would be: how comparable are the mixing state parameters chi inferred using these two alternative methods (for a subset of the 2D-PDF accessible to both methods)? – You would have the data set to look into this if you wish to do so.

19. P10, L1: The statement "Where $N_i$ is the number concentration for rBC-containing particle $i$" does not really make sense.

20. Figure 6: In the second to last minor comment made just above I suggested to compare the chi values resulting from the coupled CPMA-SP2 and SP2 only approaches. It is great to see that a similar comparison is done for the bulk MR in Fig. 6 and associated discussion. However, no information is provided whether the $m_p$ and $m_{rBC}$ ranges considered for this were constrained to the jointly accessible ranges or whether the MR from the two approaches are based on different $m_p$ and $m_{rBC}$ ranges. This kind of information is required to put the level of (dis-)agreement between the two in proper context. The same remark applies to Figs. 7 & 8.

21. P12, L10: The statement "…while the bulk MR metric cannot be used to predict the rBC mixing state in summer…" should probably read "…to predict the BC mixing state diversity index…". (bulk MR is a metric for BC mixing state)

22. P12, L12: The statement "…the rBC-containing particles were still well mixed." should probably rather be about the "mixing state diversity".

23. P12, L18: "…with the source apportionment study that about 64% of rBC-containing particles were from primary sources in winter experimental period…" – What are secondary sources of BC-containing particles!?

24. P12, L20: "…rBC-containing particles may have coatings externally mixed with the rBC cores…" – What is an externally mixed coating!?

25. P12, L26: "In summer, the primary emissions are less compared to winter, and SOA contributes more in Beijing" – In absolute or relative terms?

26. P12, L30: "However, this does not mean the amount of coatings on BC will be necessarily higher in summer because the higher temperatures and enhanced dilution may cause the primary semi-volatiles to favour the gas phase." – What would you conclude based on your study: is the fractional contribution of primary non- and semi-volatiles ever substantial when MR is high, be it winter or summer? Do we at all care about the phase partitioning of primary semi-volatiles when it comes to BC mixing state?

27. P13, L1: "The results here suggest that the increase of coating content such as the bulk MR above 4 could importantly increase the internal mixing state of rBC-containing particles" – Please choose a more precise wording. Do you rather refer to the mixing state diversity?

28. Section 5: The general importance of know BC mixing state including mixing state diversity on single particle level is motivated. This study provides likely unprecedented mixing state diversity information for BC particles. However, the important question whether the observed diversity is at a level where errors made in different applications by assuming an average mixing state are small or large is unfortunately not address. This would be desirable, though possibly a topic to be follow-up in future studies. As is, the current Section 5 could just as well be moved to the introduction as motivation for doing such measurements.

29. P14, L12: "…and makes the bulk MR may act as a predictor of mixing state…"
    a. Requires language editing.
    b. Do you mean: "…a predictor of mixing state diversity"?

30. P14, L15: "The slightly higher χ in summer indicates that internal mixing is preferred for rBC particles…" – Higher chi – for BC particles only – does not imply internal mixing of BC. Instead it says something about the particle-to-particle variability of MR.

31. Supplement, Fig. S1: it would be instructive to point out at least the obviously multiply charged particles in some way. In fact, having a full inversion scheme at hand would very easily allow to split the measured 2D-PDF into contributions from singly, doubly and triply charged particles.

**Technical comments:**
32. "Coating thickness" should be explicitly defined as it is a purely operational definition rather than being meaningful from a morphology point of view. Specifically for this study: "coating thickness" refers to a "concentric spheres equivalent coating thickness" derived from direct

measurements of total particle and BC core mass (and assumed BC and coating densities). Or would it actually be possible to largely omit the term "coating thickness"? "BC mixing state" would for example seem more appropriate in the subsection title 4.3 and the first sentence of this subsection because it is all about the "BC mixing state expressed with the non-BC to BC mass ratio of the BC-containing particles".

33. P1, L12 (abstract): repetition of "in the atmosphere"
34. P1, L30 (abstract): requires language editing (missing "to"; adjective vs adverb)
35. P2, L2 (abstract): Starting the sentence on this line with "This…" does not appear to be appropriate.
36. P2, L6: Should better read: "…component of atmospheric particulate matter and…"
37. P3, L13: "a polluted"
38. P4, L1: Please provide the year when the field experiments were done.
39. P4, L26-29: Please fix this sentence.
40. P5, L28: Please fix this sentence.
41. P5, L30: Using "$N_{BC}$" instead of solely "$N$", would be more precise as the 2D-PDF exclusively includes BC-containing particles, whereas BC-free particles are not considered.
42. P6, L3-4: It should probably read "dlog$m_{p}$" and dlog$m_{rBC}$", however, I am not a mathematician.
43. P6, L5-6: A more precise wording would include "inverted" and "retrieved from the measured…"
44. P6, L13-15: Please fix this sentence.
45. P8, L4: "approximate" instead of "appropriate"?
46. Figure 7 (and several of the following time series graphs): It would be helpful to indicate the different periods ("heavy pollution", etc.).
47. P9, L9-10: Please fix this sentence.
48. The caption of Figure 9 appears to be a revolutionary evolution.
49. Figure 9: What is included in the "MR=0" class? Everything with MR less than a small but non-zero value? Or in other words: Finite CPMA transfer function and uncertainties of CPMA and SP2 make it impossibly to accurately quantify MR=0 (even if such particles were truly existing). How is this dealt with?
50. P11, L4: The sentence "As the CPMA-SP2 system only detects the rBC-containing particles and the number of species set here is 2, i.e. rBC and non-rBC material." is incomplete. Please fix.
51. P14, L10: "…, which illustrated there was more internal mixing during this period…" is obsolete as the complete and more precise statements follow in the next sentence.
52. Supplement, Eq. 9: the superscript for the parameter $p_i^?$ appears to be missing.

**References:**

Schroder, J. C., Hanna, S. J., Modini, R. L., Corrigan, A. L., Kreidenweis, S. M., Macdonald, A. M., Noone, K. J., Russell, L. M., Leaitch, W. R., and Bertram, A. K.: Size-resolved observations of refractory black carbon particles in cloud droplets at a marine boundary layer site, Atmos. Chem. Phys., 15, 1367–1383, https://doi.org/10.5194/acp-15-1367-2015, 2015.

Motos, G., Schmale, J., Corbin, J. C., Zanatta, M., Baltensperger, U., and Gysel-Beer, M.: Droplet activation behaviour of atmospheric black carbon particles in fog as a function of their size and mixing state, Atmos. Chem. Phys., 19, 2183–2207, https://doi.org/10.5194/acp-19-2183-2019, 2019.

---

## Author Comment (AC1) · 27 Nov 2019

**Response to reviewers' comments**

Firstly, we would like to thank both referees for their important comments, we have addressed all the comments below. The original comments from referees are in black, our replies are in blue and the changes in original manuscript are in red.

Before beginning a point by point rebuttal, we feel we must clear up a specific point regarding to the "mixing state index" or the "diversity metrics" used in our study. We would like to kindly point out that Referee #3 has wrongly stated the "mixing state index" as "number-weighted", and we feel that this misunderstanding has given rise to some of their major criticisms.

We would like to state here that the "diversity metrics" used in our study are "mass-weighted", as acknowledged by Referee #1's comments. This is following from previous publications such as Riemer and West (2013) and Hughes et al. (2018), who define "mixing state index" by "mass-weighting". The parameter "bulk diversity" is also mass-weighted, consistent with Healy et al. (2014) that "bulk diversity". We recognise that we should have stated this explicitly to avoid confusion and have done so in the revised version.

**Anonymous Referee #1**

**General comments:**
This paper presents results from the APHH campaign in Beijing where mixing state of black carbon-containing particles was experimentally determined using a combination of a centrifugal mass analyzer and a single particle soot photometer. To quantify "black carbon mixing state", 2D number distributions were produced, the mass ratio of non-BC and BC material was used, as well as the mixing state parameter chi. This paper represents an important step forward in deriving mass-fraction-based metrics from single-particle measurements. I commend the authors for accomplishing this, as it is a key prerequisite for eventually being able to compare these measurements to mixing-state-aware models. The data analysis reveals interesting and novel insight into the seasonable variability of black carbon mixing state in a polluted environment such as Beijing. Overall this work warrants publication in ACP, but before it can be accepted, several issues need to be addressed. I have four major comments regarding the presentation and interpretation of the results, and a number of minor comments, mainly pertaining to the clarity of notation and terminology.

We thank the referee for the positive comments.

**Major comments:**
1. The usage of the mixing state chi is confusing in the paper. Due to the measurement technique, it is only the mixing state of BC-containing particles that can be evaluated, i.e. the method is blind to a potentially existing subpopulation that only contains nonBC material. This is an important point, and it is explained in the paper, but it comes too late. I recommend introducing the specific use of chi already in the introduction, following the overview of previous studies that used chi as a metric for mixing state (note that each of those studies have their own take on chi). The authors might consider calling their chi "χBC" or similar, to emphasize this point.

We thank the referee for the suggestion of "chi" using in our study. We have addressed this issue and have included corrections in our manuscript to make the "chi" used in our study more clear. We have changed all the terms in our manuscript into $\chi_{rBC}$, and we have added in the introduction:

In this study, the mass-weighted mixing state index for rBC-containing particles ($\chi_{rBC}$) is derived to investigate the distribution of non-rBC material across the rBC-containing particle population in Beijing.

2. Related to comment 1, the interpretation of chi is also inconsistent throughout this paper. In the abstract and a few times in the text, it is correctly mentioned that chi (as defined here) "quantifies whether the coating is evenly distributed across the rBCcontaining particle population". – I agree with this interpretation, and suggest that the authors explain this in the text more precisely, and stick to this interpretation throughout the text. I.e. a high value of chi means that all particles have the a very similar BC mass fractions (the same mass fraction for chi=100%) compared to a low value, where the distribution of BC mass fractions is very uneven across the population. I strongly suggest not to use language like "A decrease of chi was observed [. . .] which indicates a

period of more externally mixed rBC" (p. 11, line 8). The phrase "Externally mixed" is not meaningful here. In fact, I believe you can describe your results without even mentioning the terms internally and externally mixed.

We thank the reviewer for this and have revised the part that relates to this issue. We have established that a higher $\chi_{rBC}$ value represents a good mixture for rBC-containing particles where all rBC-containing particles have the similar fraction of rBC and non-rBC material. Conversely, the lower $\chi_{rBC}$ value represents a relatively uneven distribution of rBC and non-rBC material.

3. Having established the meaning of chi in your study, it is important to recognize that the diversity metrics as used here are mass-weighted. Therefore, the overall chi for the whole population will represent mainly the conditions for the large particles. It would be interesting to see how chi varies between small and large particles. Is it possible to calculate a size-resolved chi?

Thank you for the suggestion. We have added parts for the analysis of $\chi_{rBC}$ under smaller rBC-containing particle mass levels (at $m_{\mathrm{p}} \leq 15$ fg and at $m_{\mathrm{p}} \leq 5$ fg).

To investigate the mixing state of smaller rBC-containing particles, the mixing state index at smaller $m_{\mathrm{p}}$ is also calculated, shown in Figure 10. The $\chi_{rBC}$ at $m_{\mathrm{p}} \leq 15$ fg and $m_{\mathrm{p}} \leq 5$ fg is shown to be higher than the bulk $\chi_{rBC}$, indicating that the rBC was more evenly distributed among rBC-containing particles at smaller sizes. For most of the winter period, the trend of $\chi_{rBC}$ at $m_{\mathrm{p}} \leq 15$ fg and $m_{\mathrm{p}} \leq 5$ fg was similar to the trend of bulk $\chi_{rBC}$. However, during the heavy pollution period between 3rd Dec and 4th Dec, the $\chi_{rBC}$ at $m_{\mathrm{p}} \leq 5$ fg decreased slightly suggesting that there was a relatively uneven distribution of rBC material and non-rBC material for the smaller rBC-containing particles. During summer, the $\chi_{rBC}$ at $m_{\mathrm{p}} \leq 15$ fg and $m_{\mathrm{p}} \leq 5$ fg was closer to the bulk $\chi_{rBC}$ because of rBC-containing particles in summer being smaller and most of the rBC-containing particles having smaller $MR_i$.

[Figure]

(a)

[Figure]

(b)

4. Use of the terminology "core-coating". The authors emphasize that their measurement method is morphology-independent, but still frequently use the terms "BC core" and "coating". I think these terms should be avoided because it invokes the traditional BC core-shell assumption, which may or may not apply in reality (and which is actually not made by the authors!).

Accepted. We have changed the term "rBC core" to "rBC material", and "non-rBC coating" to "non-rBC material" in the context. Combined with the technical comments from Referee #3, we specified that the term "coating thickness" used in our study relates to the difference in radius of hypothetical concentric spheres of equivalent volumes.

**Minor comments:**

1. The abstract is too long. I recommend shortening it.

Accepted. Now the abstract reads as follows:

**Abstract.** Refractory Black Carbon (rBC) in the atmosphere is known for its significant impacts on climate. The relationship between the microphysical and optical properties of rBC remains poorly understood and is influenced by its size and mixing state. Mixing state also influences its cloud scavenging potential and thus atmospheric lifetime. This study presents a coupling of a centrifugal particle mass analyser (CPMA) and a single particle soot photometer (SP2) for the morphology-independent quantification of the mixing state of rBC-containing particles, used in the urban site of Beijing as part of the Air Pollution and Human Health-Beijing (APHH-Beijing) project during winter (10$^{th}$ Nov – 10$^{th}$ Dec, 2016) and summer (18$^{th}$ May – 25$^{th}$ June, 2017). This represents a highly dynamic polluted environment with a wide variety of conditions, that could be considered representative of megacity area sources in Asia. An inversion method (used for the first time on atmospheric aerosols) is applied to the measurements to present two-variable distributions of both rBC mass and total mass of rBC-containing particles and calculate the mass-resolved mixing state of rBC-containing particles, using previously published metrics. The mass ratio between non-rBC material and rBC material ($MR$) is calculated to determine the thickness of a hypothetical coating if the rBC and other material followed a concentric sphere model (the 'equivalent coating thickness'). The bulk MR ($MR_{bulk}$) was found to vary between 2 - 12 in winter and between 2 -3 in summer. This mass-resolved mixing state is used to derive the mass-weighted mixing state index for the rBC-containing particles ($\chi_{rBC}$). $\chi_{rBC}$ quantifies how uniformly the non-rBC material is distributed across the rBC-containing particle population, with 100% representing uniform mixing. The $\chi_{rBC}$ in Beijing varied 55% and 70% in winter depending on the dominant air masses and $\chi_{rBC}$ was highly correlated with increased $MR_{bulk}$ and PM$_1$ mass concentration in winter, whereas $\chi_{rBC}$ in summer varied

significantly (ranging 60% - 75%) within the narrowly-distributed $MR_{\text{bulk}}$ and was found to be independent of air mass sources. In some model treatments, it is assumed that more atmospheric ageing causes the BC to tend towards a more homogeneous mixture, but this leads to the conclusion that the $MR_{\text{bulk}}$ may only act as a predictor of $\chi_{rBC}$ in winter. The particle morphology-independent and mass-based information on BC mixing used in this and future studies can be applied to mixing-state aware models investigating atmospheric rBC aging.

2. p. 1, line 17: Which year was the campaign in?

Accepted. We have added the year as shown below:

The measurements were conducted during the APHH winter (10th Nov – 10th Dec, 2016) and summer (18th May – 25th Jun, 2017) intensive observation periods at the Institute of Atmospheric Physics (IAP) tower site.

3. p. 1, line 25: "positively associated" - Do you mean "correlated"?

Accepted.

4. p. 2, line 9: Reference to Ramanathan and Carmichael (2008) is more than 10 years. Suggest to replace with a more recent reference.

Accepted. We have added Bond et al. (2013) and Riemer et al. (2019).

5. p. 3, line 20: "measure the different mass ratio" – Should read "measure the mass ratio"?

Accepted.

Liu et al. (2017) first introduced this morphology-independent instrument configuration to measure the mass ratio between non-rBC material and rBC material.

6. p. 4, line 25: The size ranges for the detection limits are confusion. Do you mean 200-720 nm for the mixed particles (i.e. referring to the scattering signal) and 70-850 nm for the BC-component (traditionally called the BC core)? Please clarify. In this context, Figure 5 shows that the distributions extend to smaller diameters than 200 nm/70 nm. Do you also extrapolate towards the small sizes (not only the large sizes)?

We have revised the description of SP2. To be clear, we haven't extrapolated toward the small sizes. As the SP2 is only used to detect the rBC-component, we haven't applied any scattering signal data from the SP2 for the CPMA-SP2 system, as the mass provided by the CPMA will be more accurate and precise. Figure 5 contains an error which is now fixed; the Dp and Dc size is starting from ~70 nm.

The description for SP2 is revised as follows:

The SP2 can detect particles with an rBC component equivalent to a spherical diameter of 70-850 nm (Liu et al., 2010;Adachi et al., 2016).

Figure 5 has been revised as follows:

[Figure]

(a)                                                     (b)

7. p. 5, section 2.5: This is beyond the scope of this paper, but possibly an avenue for future research: Since you know the size distribution of the non-refractory particle matter from the AMS, and you know how much NR material is associated with BC, can you infer the size distribution of the subpopulation that does not contain BC by combining the AMS with the CPMA-SP2 measurements? This would yield a complete picture of aerosol mixing state.

We thank the referee for the suggestions about the future extension of our study. We have tried the combination of CPMA and AMS in our lab, but the results from AMS are not acceptable due to its lower sensitivity and the high weighting of multiply charged particles.

8. p. 6, line 22: This is an example where Major comment 2 applies. The terms internal/external mixing are not useful here. Chi as applied to the subpopulation of BC containing particles quantifies how evenly the BC and non-BC mass fractions are distributed. (with a focus on the large particles).

Accepted. We have revised this section:

The mass-weighted parameter $\chi_{rBC}$ enables the precise quantification of the distribution of rBC material and non-rBC material across the rBC-containing particle population. For a well-mixed situation (all the rBC-containing particles contain the same mass fraction of rBC material and non-rBC material), $D_\gamma = D_\propto = 2$, and the mixing state index $\chi_{rBC} = 100\,\%$. For a fully externally mixed situation (all the rBC-containing particles only contain the rBC material), $D_i = D_\propto = 1$, and the mixing state index $\chi_{rBC}$ will be undefined.

9. p. 8, line 16: "For better quantification" – what does that mean? To reduce noise?

Accepted. We have revised the sentence:

To improve the signal-to-noise ratio of the data, the measured CPMA-SP2 data have been averaged over 3-hour periods.

10. p. 9, line 1: "It would appear that .. " – can you please clarify? Do you mean that the summer distribution is similar to winter/moderate or winter/light pollution, or that it is not similar?

Yes, we are trying to say that the summer distribution is similar to the distribution in winter the moderate or light pollution period in winter. We have refined the sentence:

It appears that the $\frac{\partial^2 N_{rBC}}{\partial logm_p \partial logm_{rBC}}$ distribution in summer is similar to the distribution for moderate or light pollution periods in winter.

11. p. 9: equation 6 and 7: Suggest finding more succinct variable names for "single particle MR" and "bulk MR". Even MRi and MRbulk would be an improvement here.

Accepted. We have revised the terms used to describe different "MR".

Now the bulk MR for the rBC-containing particles population is referred as " $MR_{bulk}$ ". The single particle MR is referred as " $MR_i$ ".

12. p. 9, equation 7: Since mp,i refers to an individual particle, and the summation is over all particles, there should not be the factor Ni. In other words, isn't Ni = 1?

We have fixed the description for Bulk MR calculation as follows:

$$MR_{bulk} = \frac{\int \frac{dM_p}{dlogm_p}dlogm_p}{\int \frac{dM_{rBC}}{dlogm_{rBC}}dlogm_{rBC}} = \frac{\iint \frac{\partial^2 N_{rBC}}{\partial logm_p \partial logm_{rBC}}m_p dlogm_{rBC}dlogm_p}{\iint \frac{\partial^2 N_{rBC}}{\partial logm_p \partial logm_{rBC}}m_{rBC}dlogm_{rBC}dlogm_p} - 1 \quad (1)$$

where $M_p$ and $M_{rBC}$ are the total mass concentration for rBC-containing particle and rBC particle respectively.

13. p. 10, line 8: "due to the presence of large numbers of large rBC-particles : : : the bulk MR value increased rapidly." This seems the wrong way round, since mBC;i is in the denominator for Bulk MR.

We have fixed the discussion as follows:

As indicated by the two-variable distributions during this haze period in the previous section, there were large numbers of large rBC-containing particles with their $m_{rBC}$ between 1 fg and 10 fg. Influenced by processes such as more active condensation and coagulation during the haze period, the non-rBC species were mixed with the rBC material more effectively, and the $MR_{bulk}$ value increased rapidly.

14. p. 10, line 27: should be Figure 9. Is there anything that could be gained by comparing the bulk MR with the population average of the single particle MR?

We thank the referee for the suggestion. However, as shown in Figure 5(b), there may still be numbers of tiny rBC particles smaller than ~70 nm which are below the SP2 lower detection range and we should be careful about this. We think it may not be appropriate to work out the population average of the single particle MR here.

[Figure]

(b)

15. p. 10: Figure 9: What does MrBC mean? Caption: Should read "evolution".

We have fixed the axis label and caption. It should be $m_{rBC}$ mass fraction:

[Figure]

16. p. 10, line 32: Can you please add a reference, why coal burning should lead to thickly coated rBC-containing particles?

We have rewritten the discussion here, as the discussion may be incomplete in our original manuscript. We have revised the discussion as follows:

Figure 4 showed that during the highly polluted period, there were large numbers of rBC-containing particles with $m_{rBC}$ between 1 fg and 10 fg. These rBC-containing particles with large $m_{rBC}$ mostly originated from the southern Plateau based on the back-trajectory analysis in Fig. 10a, and Liu et al. (2018) suggested that they may be from coal burning.

17. p. 11: Comparing to Healy et al. (2014), it would be important to note that they defined chi based on the 5 species system (SO4, NO3, NH4, OA, BC). So those chi values and the one from this study are not directly comparable.

We agree that the "chi" value in this study is different from other studies. To refine this, we just introduced the conclusions from Healy et al. (2014) instead of comparing the "chi" results, and we have declared that the results from Healy et al. (2014) are based on bulk aerosol measurements.

This evolution during the winter period agrees with the findings from Paris of Healy et al. (2014), who found that when considering the mixing state of bulk aerosol mass (which considered multiple non-rBC species) there was a trend towards more homogenous mixing when polluted air masses dominate compared to the local and clean marine air mass periods.

18. p. 11: Figure 10, top panel: Specify that it's bulk diversity for the y label.

Accepted.

19. p. 12, line 6: "there was a strong correlation: : :" Can you quantify this more precisely?

Accepted. We have revised as follows:

There was a positive correlation between the variation of $\chi_{rBC}$ and the variation of $MR_{bulk}$

20. p. 12 Discussions: This section is also a good example that should be rephrased in light of Major comment 2. I think the main point of figure 12 is that during summer, for a narrow range of bulk MR a wide range of chi is observed, meaning even for similar bulk MR, there is a variation how the "coating material" is distributed (note that in summer chi is not always high). During winter, there seems to be a positive correlation between bulk MR and chi, meaning that the NR material is more evenly distributed when bulk MR is high. The explanation of why this summer/winter difference happens, however, is not clear. It would be important to improve this discussion.

We thank the referee for the comments about the discussion section. We have revised it following Major comment 2:

[revised manuscript text omitted]

21. p. 13: Implications: I appreciate the authors effort to connect their results to the modelling community, but The statement "it may be appropriate to assume internal mixing of BC in the summer" is over-reaching in my opinion. It would at least require an estimate of calculating absorption enhancement for the conditions shown here to substantiate. Water uptake (which will be modulated by the composition of the NR material) further impacts this impact. I think an important contribution of this study is that it provides a framework to produce the 2D distributions as shown in Figure 4 (mass-based, including an objective method of extrapolation). This is data that mixing-state aware models (including, but not limited to particle-resolved models) can compare to.

We thank the referee for the suggestions about our Implication parts. We have combined two referees' comments and refined this section as:

5 Implications for the future studies

The CPMA-SP2 together with a new inversion method are capable of exploring ambient BC mixing state in a way not previously possible. The detailed two variable distribution is able to retrieve the mixing state information on a single particle level. For future studies, the detailed two variable distribution results from the CPMA-SP2 can be directly compared with mixing-state aware models such as GLOMAP (Mann et al., 2010) and the particle resolved model PartMC-MOSAIC (Riemer et al., 2009) when simulating the behaviour of BC in polluted plumes. Some modelling studies have suggested that large uncertainties of BC atmospheric lifetime result from uncertainties in the scavenging efficiency (e.g. Myhre and Samset (2015)). The processes of activation and impaction scavenging are partly dictated by the overall size and soluble fraction of the rBC-containing particles, which would manifest in the data presented here. However, while some models carry this information (e.g. 'hydrophobic' vs 'hydrophilic' BC in GLOMAP) this is poorly constrained partly because of the lack of observations that can resolve this (Taylor et al., 2014). In general, more detailed non-rBC material information (MR results) and single-particle level mixing state information presented in our study can contribute to future studies concerning BC lifetime and transportation to help to constrain the simulations (Bond et al., 2013;Fierce et al., 2017).

22. Axis labels are generally too small

Accepted. We have modified the axis labels for all the graphs.

**Anonymous Referee #3**

**General comments:**
The introduction provides a good overview on reasons why a better understanding of the mixing state of atmospheric black carbon (BC) particles with other particulate matter is required when it comes to e.g. their climate impacts through aerosol-radiation and aerosol-cloud interactions. This prepares the ground for the single main topic of this study.

In a recent lab study, the combination of a coquette particle mass analyzer (CPMA) and a single particle soot photometer (SP2) was shown to be a useful and reliable means to quantify the 2-dimensional mixing state distribution as a function of total particle mass and BC core mass, which also works for complex BC aerosols with high mixing state diversity on a single particle level. Furthermore, such measurements were not accessible with comparable data coverage and accuracy to other previously applied experimental methods. The study here takes the combined CPMA-SP2 approach for the first time to the field, specifically to an air pollution hotspot where the environmental impacts of aerosols are of great interest. This certainly makes the resulting data set on BC mixing state and mixing state diversity on single particle level worth publishing.

We thank the referee for the positive comments for the scientific signification of our study.

Reasons for this judgment are given in below comments and include:
• Very imprecise and sometimes wrong statements concerning the mixing state index "chi", which is the very central topic of this study.

We agree that "Chi" is our central topic but perhaps the referee has misunderstood our use of the term. To avoid further misunderstanding, we have refined all the parts that related to the term "Chi". We have combined two referees' comments and correct all the terms in our study. In our manuscript, we have defined the "Chi" used in our study as the parameter to quantify whether the fraction of non-rBC material relative to BC material is evenly distributed across the rBC-containing particle population by mass. Now we have refined the discussion about the meaning of "Chi" results to make it clear.

• Insufficient sensitivity analyses and discussion on how the results for the mixing state index "chi" may be affected by the detection limits of the applied instruments.

We appreciate the referee's query about our measurement range. But as stated previously, the referee incorrectly states that the mixing state index is "number weighted" and we feel that gives rise to undue criticisms (reflected in Major Comment NO.2 and NO.3).

As defined in Riemer and West (2013), the mixing state index "Chi" is defined by mass-weighting, as with other published studies (i.e. Hughes et al. (2018)). Therefore, our results based on mass will not be significantly influenced by the very small particles existing below the lower detection limit of the instruments.

Referee #1's Major Comment NO.3 and Minor Comment NO.8 recognise that "Chi" is a mass-weighted parameter. As pointed out by Referee #1, "Chi" (or "Chi_rBC" specific for our study) applied here is a parameter that is weighted towards relatively large rBC-containing particles. To avoid further misunderstandings, we have emphasised that "Chi" is mass weighted in the method section:

The mass-weighted parameter $\chi_{rBC}$ enables the precise quantification of the distribution of rBC material and non-rBC material across the rBC-containing particle population

• Week discussion and interpretation of the results.

The discussion and interpretation have been improved in light of both of the reviewers' helpful and constructive comments.

• Poor language at many instances, definitely below the level that can be expected from joint forces of the whole author list.

Following both the two referees' comments, we have improved the use of language throughout.

**Major comments:**

1. P6, L24: I am puzzled about the statement "In this study, because only the rBC-containing particles are detected, it is not possible to detect a completely external mixture." – I would expect that the formalism was spitting out chi=0 (indicating completely external mixture for the particle types under consideration), if the coupled CPMA-SP2 measurements would provide m_rBC = m_p for every single particle (or more precisely: when calculating the limes towards this mixing state as the equations contain multiplication of "zero times infinity" or division of "zero by zero" for a perfectly externally mixed population).

The referee has misunderstood our use of the chi parameter. What we are trying to explain is that the calculation of mixing state index will not return a fully externally mixed result (with $\chi = 0$) for our rBC-containing particles. Because the division of "zero by zero" will leave the $\chi$ results undefined rather than 0. We have revised as follows:

For a fully externally mixed situation (all the rBC-containing particles only contain the rBC material), $D_i = D_\alpha = 1$, and the mixing state index $\chi_{rBC}$ will be undefined.

Furthermore, according to the CPMA-SP2 inversion results from the chamber experiment presented by Broda et al. (2018), the uncoated, bare rBC-containing particles can be measured and presented.

2. Truncation of the BC particle mixing state 2D-PDF due to detection limits of the applied instruments is often unavoidable, and so it is in this study. To my judgement, the authors have done a fair job in addressing the impact on the resulting mixing state parameter from the upper limit of measured total particle mass (at least for 3 out of the 4 examples; separate comment on the latter follows). However, they have not at all addressed the impact of the SP2 lower detection limit for rBC mass in a single particle. The 2D-PDFs very nicely illustrate that for total particle mass equal to the SP2 lower detection limit only externally mixed BC particles can be measured, while internally mixed BC with, consequently, smaller BC core remains undetected. From a pessimistic perspective, the nearly complete mixing state information for BC containing particles is only available for either a narrow range of BC core sizes (roughly indicated by red square), or for a narrow range of total particle mass (roughly indicated by the magenta square).

How does the overall mixing state diversity parameter compare with its value calculated for the above two constrained size ranges? Does this hamper absolute interpretation of the chi values or is this only a minor issue? Personally, I do not really have a feeling for the sensitivity of chi to such truncation effects. However, the fact that chi appears to be weighted by particle number and that particle number is dominated by small particles in the range of the lower SP2 cut-off would suggest that truncation at the lower SP2 cut-off could have a much more dramatic effect on the resulting chi value than the truncation at the upper CPMA cut-off in many cases (the heavy pollution period is obviously the counter-example).

As mentioned in the earlier general response, the comment "the fact that chi appears to be weighted by particle number and that particle number is dominated by small particles" is incorrect due to "Chi" being mass-weighted. If the referee is referring to the "the overall mixing state diversity parameter" to "bulk diversity $D_\alpha$", it is described in Healy et al. (2014) that "bulk diversity $D_\alpha$" is also mass weighted. We agree with the reviewer that at the lower detection limitation of SP2 the number concentration is high, but these rBC particles contribute little to the total rBC mass concentration as shown in the figure below:

[Figure]

In addition, if we do the extrapolation to the lower limitation of the SP2, there will not be any effective method for us to examine this extrapolation; unlike the upper mass limit (which is dictated by the highest setting of the CPMA), we cannot compare with the 'SP2 only' data because this suffers from the same limitation. As such, we do not see the value in performing this extrapolation.

3. Taking the previous comment a step further would raise the question whether a number-weighted parameter is suitable when it comes to e.g. BC lensing effect, where we care in first order approximation about the BC core size range where the BC mass size distribution peaks. Would it also be useful to additionally compute the particle diversity diameter exclusively for a narrowish range of rBC core mass around the modal size of the rBC mass size distribution (which will require some extrapolation above the upper CPMA cut)? This might also provide a mixing state information that is more readily comparable across different studies with similar instruments as they only have to cover this range well, whereas the actual lower/upper instrument cut-offs would loose out in relevance.

We thank the referee for the suggestions. However, following the response above, we respectfully disagree with the statement about "Chi" because it is mass weighted and therefore is already weighted towards the centre of the distribution – the part which the referee wishes us to focus on in the comment.

The "Chi" defined in this study is the mass weighted parameter to quantify whether the non-rBC material is relatively evenly distributed across the rBC-containing particle population. Therefore, we think "Chi" can be used in comparison to the other mass weighted parameters, and "Chi" will focus on the peak of mass distribution.

4. The extrapolation of the mixing state PDF looks reasonable for the example shown in the SI (and likely for the low/medium pollution winter and the summer cases). However, showing the heavy pollution period mixing state results without indicating the measured/extrapolated parts of the 2D-PDFs is really misguiding the reader: While the measured part still provides valid information on the fact the BC is generally highly internally mixed, the mixing state diversity parameter chi, which is vastly dominated by the extrapolated part of the 2D-PDF, can hardly be meaningful in absolute terms.

Following the referee's suggestion, we have added the black solid line on the inversion graphs in Figure 4 to indicate where the extrapolation starts. We agree that the mixing state parameters are dominated by large particles during the heavy pollution period, but this is from a mass weighted perspective rather than a number weighted perspective.

To validate our extrapolation method, we have discussed our CPMA-SP2 extrapolated results with the SP2 only results in the extrapolated session (Section 3.3). As the rBC mass concentration from the CPMA-SP2 'Fit MrBC' extrapolation is close to the results from 'SP2 only' method, we think this will deliver an acceptable result.

Furthermore, we think the extrapolation will not influence our conclusion drawn from the mixing sate index "chi". Following the response to the major comment NO.3 made by Referee #1, we have included an analysis of the mixing state index for a selected, smaller range of $m_p$ values. We show that the same conclusion can be

drawn for the parts where the extrapolation is not applied (as indicated by the results for mixing state index where $m_p \leq 15$ fg).

[Figure]

(a)

5. P10, L6-13: While not being wrong, the discussion in this paragraph is rather weak. In the end the observed mixing state is a result of various factors including mixing state at emission, average atmospheric residence time of the BC particles, representative aging rate (in the sense of coating acquisition rate), etc.
a. For the high pollution winter case the authors simply say: high secondary aerosol production → more condensation → more rapid coating acquisition → higher MR. This is incomplete.
b. For the low pollution winter case the authors simply say: observed thin coatings may indicate fresh emission. This is also incomplete.
c. The statement "due to the presence of large numbers of large rBC containing particles with their $mm$rBC between 1 fg and 10 fg, the bulk MR value increased rapidly" does not really make sense (BC core size alone cannot explain anything about MR).
d. Further down (P10, L30) the following statement is made for the high pollution period: "This large fraction of thickly coated rBC-containing particles was expected due to the coal burning in winter." – How does this relate to above arguments and where the coal burning BC mixing state knowledge come from?

We thank the referee for the comments. We agree that the observed mixing state of rBC-containing particles is associated with both the emissions and atmospheric process. However, for a detailed discussion for the mixing state at emission, average atmospheric residence time of the BC particles, representative aging rate (in the sense of coating acquisition rate) may be beyond the scope of our study.

We have revised the discussions for comments (a), (b) and (c):

The CPMA-SP2 $MR_{bulk}$ time series for winter and summer are presented in Figure 7. A significant increase was observed during the winter heavy pollution period, and the $MR_{bulk}$ reached a maximum of 10 on December 4th. During low pollution periods the rBC particles had a lower $MR_{bulk}$ around 2. According to the source apportionment work during the same study by Liu et al. (2018) and Wang et al. (2019), around 64% of organic components within the rBC-containing particles were from primary sources in winter. These sources may also emit non-rBC material externally mixed with rBC. During the heavy pollution period in winter, a significant enhancement of secondary organic aerosol was observed (Wang et al., 2019). Figure 4 showed that during the highly polluted period, there were large numbers of rBC-containing particles with $m_{rBC}$ between 1 fg and 10 fg. These rBC-containing particles with large $m_{rBC}$ mostly originated from the southern Plateau based on the back-trajectory analysis in Fig. 10a, and Liu et al. (2018) suggested that they may be from coal burning. High pollution tends to introduce increased concentrations of gas precursors promoting the condensation of secondary material, and the higher overall particulate concentration promotes coagulation, particularly at the top of the boundary layer where the high humidity causes particles to grow further. Both condensation and coagulation will lead to more efficient mixing of rBC and increased $MR_{bulk}$. In contrast, during the lightly polluted periods, the fractions of non-rBC material within the rBC-containing particle population were much lower. The

relatively cleaner air masses from Northern Plateau dominated most of the light pollution periods (Liu et al., 2018). With the absence of high absolute non-rBC material concentrations, the rBC-containing particles were externally mixed or with lower fraction of non-rBC material.

Figure 7 shows that the $MR_{bulk}$ in summer varied between 1 and 2.7 with several moderate $MR_{bulk}$ values ($MR_{bulk}$ >2), which was generally lower than in winter. Compared to the winter periods, the source apportionment shows that secondary material contributed more to the total organic aerosol species within the rBC-containing particles (Xie et al., 2019;Wang et al., 2019).

The revised discussion for (d) is reflected in the response to major comments NO.6.

6. Presentation of results shown in Fig. 9 (last paragraph of Sect. 4.3): The potential evolution of MR during aging after emission has strong implications on which type of conclusions can or cannot be drawn. This is not properly acknowledged in the current discussion which is, therefore, flawed.

We thank the referee for their comments. We have refined the discussion regarding to Figure 9:

The significant enhancement of the fraction of rBC-containing particles with $MR_i$ ≥3 was more likely related to the increase of the fraction of secondary components during the heavily polluted period in winter (Wang et al., 2019). The rBC-containing particles with a small fraction of non-rBC material (0<$MR_i$<1.5) accounted for a large portion (around 60%) during the lightly pollution periods in winter and occurred most of the time in summer. These rBC-containing particles with much lower $MR_i$ were likely to be generated from the consumption of fossil fuel (Liu et al., 2017;Liu et al., 2018).

7. P12, L16: The following statement is made here: "This means although the rBC mass loading was low in summer however a more homogenous distribution of coatings or internal mixing state was present." – What is the expected process that causally links BC mass concentration with BC mixing state homogeneity? (Or do you have cross-correlations between BC mass concentration and other parameters that causally relate to BC mixing state diversity in mind?). The point raised in this comment also questions the meaningfulness of Fig. 12c.

We thank the referee for the comment here. Fig 12(c) is our main point of reference for the 'general pollution level'. In Fig 12(a), the x-axis is NR-PM$_1$ rather than PM$_1$ which means the rBC-containing particle mass concentration is excluded, so we have added Fig 12(c). To make this clearer, we have combined two graphs together:

[Figure]

Figure 12 (a) The relationship between NR-PM$_1$ and rBC mass concentration and Mixing State Index, (b) the relationship between Mass Ratio (MR) and Mixing State Index

The key statement should be, "although the pollution level is lower in summer" and we have fixed this.

**Minor comments:**

8. P1, L25 (abstract): The "or" in the following statement sounds confusing: "χ of rBC-containing particles was highly positively associated with increased bulk MR, rBC mass loading or pollution level in winter,…"

We thank the referee for the comment here. Following the response to Major Comment 7, we have revised this term:

$\chi_{rBC}$ was highly correlated with increased $MR_{\text{bulk}}$ and $PM_1$ mass concentration in winter.

9. P1, L29 (abstract): The following statement is difficult to understand and hardly self-explaining without having read the manuscript: "The same level of bulk MR corresponded with a higher χ in summer than in winter and this tended to suggest a limited formation of coatings on rBC largely depended on primary sources."

We thank the referee for the comment here. We have deleted this part as it may be too early to clarify this in the abstract.

10. P1, L32 (abstract): What is the mechanism that links ambient temperature with BC coating thickness?

We have refined the statement here. Please see our response to minor comments NO.1 from Referee #1.

11. P1, L34 (abstract): The statement "The mixing state of rBC-containing particles should also depend on the coating formation mechanism, both primary source influence and secondary coating formation mechanism should be considered in interpreting the rBCcontaining particles mixing state in the atmosphere." should rather be made early in the abstract, i.e. before providing interpretations of observed BC mixing state degree and variations.

We thank the referee for the comment here. We have deleted that statement in the Abstract. (see our response to minor comments NO.1 from Referee #1)

12. P2, L18-21: the Shiraiwa 2007 study cited here in the context of relationship between BC mixing state and its droplet activation does not provide any activation measurements. Schroder et al. (2015) or Motos et al. (2019) are examples of recent studies which directly investigated the activation of BC particles in ambient clouds as a function of their size and mixing state.

Accepted. We have replaced the references cited here.

13. Sect. 2.4 on CPMA-SP2 coupled system: Are small uncharged particles passing the CPMA in addition to the charged and mass selected particles an issue (upper cut-off size for uncharged particles depends on rotational speed)?

The uncharged particles didn't appear to be an issue at the CPMA setpoints used during this experiment, noting that they would appear at the bottom right of the two-dimensional plots. But to be sure, we added a filter to the inversion code (based on their scattering profiles) to remove these particles during processing.

14. P8, L12-14: Measurements of major chemical components of atmospheric aerosols rarely provide "zero". Instead, concentrations can drop by several orders in magnitude or below the lower limit of quantification. Therefore it is not very useful to report observed value ranges as "from zero to ….".

Accepted.

In contrast, the rBC and NR-PM$_1$ mass concentration reduced significantly during the light pollution periods. Compared to the winter periods, the rBC mass concentration varied less significantly in summer, and the peak of rBC mass concentration and NR-PM$_1$ mass concentration is lower. The rBC mass concentration reached a peak of around 4 µg · m$^{-3}$ while the NR-PM$_1$ mass concentration reached a peak of around 100 µg · m$^{-3}$.

15. Figure 3: The period marked as "light pollution" looks like what is described as "clean" in the first paragraph of Sect. 4.1. Actually, a few lines further down (P.8, L20) I came across the definition "light pollution means NR-PM1 < 100 ug/m3", which is not congruent with the shading shown in Fig. 3.

Accepted. We have fixed this, and it should be described as "light pollution" for the period "NR-PM1 < 100 ug/m3"

16. P9, L5: Here it says: "The volume equivalent diameter for rBC-containing particles ($D$p) and the rBC core ($D$c) is also calculated by assuming a density of 1.2 g/cm3 for rBC-containing particles and a density of 1.8 g/cm3 for rBC cores." – Strictly speaking, different densities should be assumed for core and coating then it comes to estimating $D$p from mp. Furthermore, an average density of 1.2 g/cm3, which is lower than that of BC, inorganic salts and likely a fair fraction of the organics, appears to be a bit low. However, doing it more precisely would likely have little effect on the resulting Dp values.

We don't think it will be an issue, as the $D$c and $D$p values presented here just serve to estimate the size range of our study.

17. P9, L9-11: The shift of the rBC core size distribution to larger sizes appears to be quite dramatic (it would definitely look dramatic when shown as mass rather than number size distribution).
a. Are these rBC core size distributions based on the polydisperse SP2 measurements or on the (extrapolation dominated) coupled CPMA-SP2 measurements? I would only trust the latter for the high pollution period (unless that one was jeopardized by coincidence issues).
b. How do these findings on rBC core size distribution compare with existing literature (if any)?

Firstly, there is a label error for Dc axis in Figure 5 in our original manuscript (see our response to minor comments NO.6 to Referee #1). We have fixed that error, and now that shift range is correct. The mass distribution result is shown in the response to Major comment NO.2.

Secondly, for the comparison of Dc, the 'SP2 only' results are published in Liu et al. (2018), where the mass median Dc is derived.

18. Sect. 4.3: The claim is made here that the CPMA-SP2 combination more accurately provides the MR than SP2 only data (combining incandescence signals with LEO-fit based optical sizing). This claim seems fully justified. A quite interesting question would be: how comparable are the mixing state parameters chi inferred using these two alternative methods (for a subset of the 2D-PDF accessible to both methods)? – You would have the data set to look into this if you wish to do so.

We thank the referee for the suggestions, however we do not feel that this will deliver as useful data because the LEO method of optical sizing is both less accurate and less precise a measure of overall particle size and entails making assumptions concerning the coating configuration and its effect on scattering enhancement. Furthermore, the LEO fits for small particles can be unreliable, so small, thinly-coated particles may be systematically rejected and thus bias the data. As such, we are not confident that the parameters derived will be physically meaningful.

19. P10, L1: The statement "Where $N$i is the number concentration for rBC-containing particle $i$" does not really make sense.

We have fixed the description for MR calculation:

$$MR_{bulk} = \frac{\int \frac{dM_p}{dlogm_p}dlogm_p}{\int \frac{dM_{rBC}}{dlogm_{rBC}}dlogm_{rBC}} = \frac{\iint \frac{\partial^2 N_{rBC}}{\partial logm_p \partial logm_{rBC}}m_p dlogm_{rBC}dlogm_p}{\iint \frac{\partial^2 N_{rBC}}{\partial logm_p \partial logm_{rBC}}m_{rBC}dlogm_{rBC}dlogm_p} - 1 \qquad (2)$$

Where $M_p$ and $M_{rBC}$ are the total mass concentration for rBC-containing particle and rBC particle respectively.

20. Figure 6: In the second to last minor comment made just above I suggested to compare the chi values resulting from the coupled CPMA-SP2 and SP2 only approaches. It is great to see that a similar comparison is done for the bulk MR in Fig. 6 and associated discussion. However, no information is provided whether the mp and mrBC ranges considered for this were constrained to the jointly accessible ranges or whether the MR from the two approaches are based on different mp and mrBC ranges. This kind of information is required to put the level of (dis-)agreement between the two in proper context. The same remark applies to Figs. 7 & 8.

We have added the description for "SP2 only" method as follows:

The $MR_{bulk}$ from the SP2-only method is derived by applying the SP2 LEO fitting method to rBC-containing particles with $D_p$ in the range of 0.08 ~ 0.8 μm, and $D_c$ in the range of 0.08 ~ 0.55 μm.

21. P12, L10: The statement "…while the bulk MR metric cannot be used to predict the rBC mixing state in summer…" should probably read "…to predict the BC mixing state diversity index…". (bulk MR is a metric for BC mixing state)

We have refined the sentence. We think that it should be "mixing state index" to avoid confusion with the term "bulk diversity".

In order to investigate which other parameters may act as a predictor for mixing state, Figure 12 shows the variation of $\chi_{rBC}$ in both summer and winter against (a) NR-PM₁ and rBC mass concentration, and (b) $MR_{bulk}$. There was a positive correlation between the variation of $\chi_{rBC}$ and the variation of $MR_{bulk}$ in winter as rBC-containing particles with larger non-rBC material (larger $MR_{bulk}$) tended to exhibit a more even mixture during the winter campaign period. For the summer period, no such positive correlation was found. Although the rBC-containing particles did not contain a large fraction of non-rBC material in summer, and the pollution level was lower, there was significant variation of $\chi_{rBC}$ which indicates that the distribution of rBC and non-rBC material mass fractions among the rBC-containing particles varies in a narrow range of lower $MR_{bulk}$. This indicates that higher $MR_{bulk}$ can be related to increased levels of a more even mixture of rBC-containing particles in winter, while the $MR_{bulk}$ metric cannot be used to predict the variation of $\chi_{rBC}$ in summer.

22. P12, L12: The statement "…the rBC-containing particles were still well mixed." should probably rather be about the "mixing state diversity".

Following the Response to comment 21, we have fixed the sentences.

23. P12, L18: "…with the source apportionment study that about 64% of rBC-containing particles were from primary sources in winter experimental period…" – What are secondary sources of BC-containing particles!?

We have fixed the description as follows:

around 64% of organic component within rBC-containing particle was from primary sources in winter

24. P12, L20: "…rBC-containing particles may have coatings externally mixed with the rBC cores…" – What is an externally mixed coating!?

Combining two referee's comments, we have revised the discussion section. See our response to minor comments NO.20 from Referee #1.

25. P12, L26: "In summer, the primary emissions are less compared to winter, and SOA contributes more in Beijing" – In absolute or relative terms?

In relative terms. We have fixed the sentence as follows:

and SOA contributes a greater fraction of non-rBC material compared to the winter period in Beijing

26. P12, L30: "However, this does not mean the amount of coatings on BC will be necessarily higher in summer because the higher temperatures and enhanced dilution may cause the primary semi-volatiles to favour the gas phase." – What would you conclude based on your study: is the fractional contribution of primary non- and semi-volatiles ever substantial when MR is high, be it winter or summer? Do we at all care about the phase partitioning of primary semi-volatiles when it comes to BC mixing state?

We have fixed the discussion here. See our response to minor comments NO.20 from Referee #1.

27. P13, L1: "The results here suggest that the increase of coating material such as the bulk MR above 4 could importantly increase the internal mixing state of rBC-containing particles" – Please choose a more precise wording. Do you rather refer to the mixing state diversity?

We have fixed the term. See our response to minor comments NO.20 from Referee #1.

28. Section 5: The general importance of know BC mixing state including mixing state diversity on single particle level is motivated. This study provides likely unprecedented mixing state diversity information for BC particles. However, the important question whether the observed diversity is at a level where errors made in different applications by assuming an average mixing state are small or large is unfortunately not address. This would be desirable, though possibly a topic to be follow-up in future studies. As is, the current Section 5 could just as well be moved to the introduction as motivation for doing such measurements.

Combining two referees' comments we have refined the section. See our response to minor comments NO.21 from Referee #1.

29. P14, L12: "…and makes the bulk MR may act as a predictor of mixing state…"
a. Requires language editing.
b. Do you mean: "…a predictor of mixing state diversity"?

We have fixed the term as follows:

This result shows that higher pollution levels will increase the non-rBC material in rBC-containing particles and a promote more homogenous distribution of non-rBC material and rBC material in winter. These results further imply that the $MR_{\mathrm{bulk}}$ may be a good predictor of $\chi_{rBC}$ in this season.

30. P14, L15: "The slightly higher $\chi$ in summer indicates that internal mixing is preferred for rBC particles…" – Higher chi – for BC particles only – does not imply internal mixing of BC. Instead it says something about the particle-to-particle variability of MR.

We agree with the referee's comments here. We have refined description about the meaning of higher $\chi_{rBC}$ and lower $\chi_{rBC}$ in our study.

31. Supplement, Fig. S1: it would be instructive to point out at least the obviously multiply charged particles in some way. In fact, having a full inversion scheme at hand would very easily allow to split the measured 2D-PDF into contributions from singly, doubly and triply charged particles.

The removal of multiply charged particle is described in the publication Broda et al. (2018) describing the inversion method. We state that we have removed the multiply charged particles in the section of CPMA-SP2 inversion method:

The multiply charged particles are removed through the inversion process.

**Technical comments:**

32. "Coating thickness" should be explicitly defined as it is a purely operational definition rather than being meaningful from a morphology point of view. Specifically, for this study: "coating thickness" refers to a "concentric spheres equivalent coating thickness" derived from direct measurements of total particle and BC core mass (and assumed BC and coating densities). Or would it actually be possible to largely omit the term "coating thickness"? "BC mixing state" would for example seem more appropriate in the subsection title 4.3 and the first sentence of this subsection because it is all about the "BC mixing state expressed with the non-BC to BC mass ratio of the BC-containing particles".

We thank the referee for the suggestions for the "coating thickness" using here. We have changed the section name into:

Mass ratio between non-rBC material and rBC material.

And we have specified the "coating thickness" term used in our study:

The concentric spheres equivalent coating thickness information is presented through the mass ratio (MR) parameter which is derived from the CPMA-SP2 inversion results.

It would be difficult to replace all instances of the "coating thickness" term, because this term is routinely used in the studies cited in the introduction section and many articles employing the SP2 LEO fitting method. Besides that, changing the term "coating thickness" into "BC mixing state" may lead to misunderstandings in the mixing state index section.

In addition, following Referee #1's major suggestion (major comment NO.5), we have changed all the term of "rBC core" and "non-rBC coating" in our manuscript. We have also replaced the term "thickly coated" in our study.

33. P1, L12 (abstract): repetition of "in the atmosphere"

We have revised the abstract combining the two referee's comments. See our response to minor comments NO.1 from Referee #1.

34. P1, L30 (abstract): requires language editing (missing "to"; adjective vs adverb)

Please refer to our response to minor comments NO.1 from Referee #1.

35. P2, L2 (abstract): Starting the sentence on this line with "This…" does not appear to be appropriate.

Please refer to our response to minor comments NO.1 from Referee #1.

36. P2, L6: Should better read: "…component of atmospheric particulate matter and…"

Accepted.

Black Carbon (BC) is an important light absorbing carbonaceous component of the atmospheric particulate matter and is regarded as dominant amongst absorbing aerosols in the atmosphere

37. P3, L13: "a polluted"

Accepted.

38. P4, L1: Please provide the year when the field experiments were done.

Accepted.

The measurements were conducted during the APHH winter (10th Nov – 10th Dec, 2016) and summer (18th May – 25th Jun, 2017) intensive observation periods at the Institute of Atmospheric Physics (IAP) tower site.

39. P4, L26-29: Please fix this sentence.

Accepted.

The SP2 can detect particles with an rBC component equivalent to a spherical diameter of 70-850 nm (Liu et al., 2010;Adachi et al., 2016)

40. P5, L28: Please fix this sentence.

Accept.

The multiply charged particles are removed through the inversion process, and a two-variable distribution function is used to describe the distribution of the non-rBC material on the rBC particles:

41. P5, L30: Using "NBC" instead of solely "N", would be more precise as the 2D-PDF exclusively includes BC-containing particles, whereas BC-free particles are not considered.

Accepted, we have changed all the terms into $N_{\mathrm{rBC}}$.

42. P6, L3-4: It should probably read "dlogmp" and dlogmrBC", however, I am not a mathematician.

Either "$\partial$" or "d" works depending on whether it is explained from its physical meaning or mathematical meaning. We have elected to maintain consistency with Broda et al. (2018)

43. P6, L5-6: A more precise wording would include "inverted" and "retrieved from the measured…"

We have refined this sentence:

An example of the original inverted two-dimensional distribution $\frac{\partial^2 N_{\mathrm{rBC}}}{\partial \log m_{\mathrm{p}} \partial \log m_{\mathrm{rBC}}}$ graph retrieved from the measurement is presented in the Figure S2.

44. P6, L13-15: Please fix this sentence.

We have revised this sentence:

The mass fractions of rBC material and non-rBC material for each single rBC-containing particle and the bulk rBC-containing particle population that are used for the calculation are derived from the measured mass parameters.

45. P8, L4: "approximate" instead of "appropriate"?

Accepted.

the Fit $m_{\mathrm{rBC}}$ method is better to approximate the total rBC mass compared to the Fit Ratio method

46. Figure 7 (and several of the following time series graphs): It would be helpful to indicate the different periods ("heavy pollution", etc.).

We thank the referee for their suggestion. The marked area in Figure 3(a) highlights examples of different pollution types. To mark all the polluted level among the whole time-series on the graphs would make the graphs substantially less clear and harder to read so we have decided to show the examples only.

47. P9, L9-10: Please fix this sentence

The number concentration size distribution for the $m_{\mathrm{rBC}}$ ($\frac{\mathrm{d}N_{\mathrm{rBC}}}{\mathrm{dlog}m_{\mathrm{rBC}}}$) shown in Figure 5(b) indicates that there is an increase in the number concentration for the rBC-containing particles with relatively large rBC material (with $m_{\mathrm{rBC}}$ larger than 1 fg) during the heavy pollution period in winter.

48. The caption of Figure 9 appears to be a revolutionary evolution.

Fixed. See the response to minor comments NO.15 from Referee #1.

49. Figure 9: What is included in the "MR=0" class? Everything with MR less than a small but non-zero value? Or in other words: Finite CPMA transfer function and uncertainties of CPMA and SP2 make it impossibly to accurately quantify MR=0 (even if such particles were truly existing). How is this dealt with?

We have fixed the term here to $MR_i \sim 0$ which represents the group of rBC-containing particles with $MR_i$ close to 0.

50. P11, L4: The sentence "As the CPMA-SP2 system only detects the rBC-containing particles and the number of species set here is 2, i.e. rBC and non-rBC material." is incomplete. Please fix.

Accepted. We have revised as follows:

As the CPMA-SP2 system only detects the rBC-containing particles, therefore the number of species set here is 2; the rBC and non-rBC material.

51. P14, L10: "…, which illustrated there was more internal mixing during this period…" is obsolete as the complete and more precise statements follow in the next sentence.

Accepted. We have fixed the sentence:

which illustrated there was a more evenly distribution of rBC and non-rBC material during this period

52. Supplement, Eq. 9: the superscript for the parameter $pi$ appears to be missing.

There is no superscript for the parameter $pi$ in Eq. 9. We have revised the explanation to the parameters for $p_i$. As $p_i$ is the mass fraction of particle $i$ in the particle population which is also a key parameter used to calculate the mixing state index.

The average particle mixing entropy ($H_\propto$) can be expressed as:

$$H_\propto = \sum_{i=1}^{N} p_i H_i \quad (3)$$

Where $p_i$ is the mass fraction of particle $i$ in the population;

References

[revised manuscript text omitted]